# Progressive Residual Tensor Networks for Adversarial Purification

## Abstract

Adversarial perturbations remain a critical threat to modern vision systems, and tensor network–based purification is a promising direction thanks to its low-rank priors. Yet these methods face a fundamental reconstruction–denoising conflict: preserving pixel fidelity risks recovering adversarial residues, whereas aggressive suppression sacrifices semantic detail. We address this trade-off with a Laplacian pyramid–inspired framework that performs progressive residual reconstruction across scales. At coarse levels, the rank is kept relatively unconstrained to capture global semantic structure, while a monotonically decreasing rank schedule constrains capacity at finer levels, preventing the accumulation of fine-scale perturbations. In addition, we introduce a wavelet-based residual regularization that penalizes the energy of reconstructed high-frequency subbands, discouraging residuals from absorbing adversarial noise without undermining semantic recovery. Extensive experiments on CIFAR-10, CIFAR-100, and ImageNet show that our method preserves strong reconstruction quality on clean images while substantially improving robustness under diverse attacks, demonstrating the effectiveness of a frequency-aware, progressive tensor-network purifier.

## 1 Introduction

Deep learning has achieved remarkable success across domains such as computer vision, natural language processing, and autonomous systems, driven by the strong representational capacity of deep neural networks (DNNs). However, their vulnerability to adversarial attacks has raised serious security concerns. By injecting imperceptible perturbations into the input, adversaries can cause models to output incorrect predictions with high confidence (Hung-Quang et al., 2024a; Goodfellow et al., 2015), undermining the reliability of DNNs in safety-critical applications. To circumvent these limitations, adversarial purification (AP) has been explored as a model-agnostic alternative that preprocesses inputs before classification. In particular, diffusion-based AP (Pei et al., 2025; Sun et al., 2025; Hung-Quang et al., 2024b) leverages strong generative priors to achieve state-of-the-art robustness performance, demonstrating the promising future of the AP direction.

Recently, researchers have explored tensor decompositions as a structural prior for adversarial purification. Tensor networks such as Tensor-Train (TT, Oseledets, 2011) and Quantized Tensor-Train (QTT, Khoromskij, 2011) provide compact representations that capture structural regularities with far fewer parameters than the original signal (Loeschcke et al., 2024). Their inherent low-rank bias naturally suppresses spurious high-frequency variations while preserving the dominant low-frequency content of natural images. This property makes them especially appealing for adversarial purification: projecting an input onto a low-rank tensor manifold can effectively filter out adversarial perturbations without retraining the classifier (Entezari & Papalexakis, 2022; Bhattarai et al., 2023).

Nevertheless, existing tensor-based defenses face intrinsic limitations. For instance, TensorShield employs low-rank projections that effectively suppress perturbations, but this compression often comes at the cost of fine details, leading to a loss of semantic cues and thus lower classification accuracy (Entezari & Papalexakis, 2022). Conversely, coarse-to-fine tensor networks attempt to reconstruct images at multiple scales with high fidelity, but also tend to recover adversarial residues alongside semantic structures (Loeschcke et al., 2024). This reflects a fundamental reconstruction–denoising conflict: robust purification requires discarding high-frequency perturbations, whereas pixel-accurate reconstruction demands preserving them. Frequency-domain analyses fur-

ther show that adversarial perturbations and semantic textures are often entangled in similar spectral bands, making it difficult for simple low-rank approximations to selectively filter noise without damaging useful content (Pei et al., 2025). As a result, existing tensor-based methods often achieve only a suboptimal balance between robustness and fidelity, underscoring the need for designs that explicitly incorporate scale-aware priors and frequency-sensitive mechanisms.

To address this challenge, we design a Laplacian pyramid–like progressive residual reconstruction framework that decomposes image recovery into frequency bands: each stage reconstructs only the residual at its resolution while inheriting lower-frequency content from coarser stages, ensuring that the "clean" component of each band is learned and avoiding amplification of perturbations across scales. In addition, we introduce a lightweight wavelet-based residual regularization that penalizes the energy of Haar high-frequency subbands. This design constrains the residual channels during reconstruction, preventing overfitting to unstable details and improving the balance between fidelity and robustness. Finally, we employ resolution-dependent rank scheduling, gradually constraining tensor-network capacity as resolution increases. Together, residual decomposition, wavelet regularization, and adaptive rank control partially decouple reconstruction from denoising and yield a more balanced trade-off between fidelity and robustness (Xu et al., 2018; Lai et al., 2018).

We empirically evaluate the performance of our method by comparing it with AT and AP methods across attack settings using multiple classifier models on CIFAR-10, CIFAR-100, and ImageNet. On CIFAR-10, combining our purifier with a robust classifier reaches 73.04% ($\ell_\infty$) / 81.25% ($\ell_2$) robust accuracy; on ImageNet, it achieves 48.44% ($\ell_\infty$), while the standalone purifier maintains strong reconstruction quality and competitive robustness among AP baselines.

In summary, our contributions are as follows:

- We propose a Laplacian pyramid–inspired architecture that progressively reconstructs images across frequency bands. By isolating residuals at each scale, this design alleviates the inherent conflict between faithful reconstruction and noise suppression.
- To further mitigate reconstruction–denoising conflict, we combine resolution-dependent rank scheduling with a lightweight wavelet penalty on high-frequency subbands. These complementary constraints reduce residual overfitting and enhance robustness while preserving essential semantics.
- We conduct comprehensive experiments on CIFAR-10, CIFAR-100, and ImageNet under various attack settings and classifier backbones, demonstrating that our method significantly improves adversarial robustness while maintaining competitive clean accuracy compared to state-of-the-art AT and AP approaches.

## 2 RELATED WORK

### 2.1 ADVERSARIAL ROBUSTNESS

Adversarial attacks continue to challenge the reliable deployment of deep neural networks. The most established defense, adversarial training (AT), casts robustness as a min–max problem and augments training with adversarial examples; despite its effectiveness, AT is computationally demanding, suffers from robust overfitting, and often trades clean accuracy for robustness (Robey et al., 2023; Wu et al., 2024; Gowda et al., 2024). Fairness-aware studies also report that AT may amplify group disparities without distance-aware objectives (Lee et al., 2024). As a model-agnostic alternative, adversarial purification (AP) preprocesses inputs prior to classification. Diffusion-based AP leverages strong generative priors but is sensitive to noise schedules and sampling steps and may remove not only perturbations but also useful high-frequency details; frequency-domain analyses show that adversarial perturbations overlap with semantic textures, complicating selective suppression (Pei et al., 2025; Sun et al., 2025; Hung-Quang et al., 2024b). These limitations motivate the use of structural priors that can better separate perturbations from essential image content.

### 2.2 TENSOR DECOMPOSITION AND ADVERSARIAL PURFICATION

Tensor networks (TNs) such as Tensor-Train (TT, Oseledets, 2011) and Quantized Tensor-Train (QTT, Khoromskij, 2011) offer compact factorizations that capture the structural information of

images with far fewer parameters than the original representation (Loeschcke et al., 2024). Their inherent low-rank bias suppresses undesired high-frequency fluctuations while preserving dominant low–mid-frequency content, which makes TNs natural candidates for purification. Early work TensorShield projects inputs onto a low-rank tensor subspace, filtering a substantial portion of adversarial perturbations but at the cost of discarding fine details and thus leading to a loss of semantic cues and thus lower classification accuracy (Entezari & Papalexakis, 2022). Subsequent factorization-based defenses further validate that low-rank structure improves robustness, yet aggressive compression induces a fidelity–robustness trade-off (Bhattarai et al., 2023). Building on coarse-to-fine training of TNs for image reconstruction (Loeschcke et al., 2024), recent model-free AP (TNP) integrates a coarse-to-fine prior by using the lower-resolution reconstruction as a clean prior constraint for the next scale (Lin et al., 2025). However, this overly strong prior can force away semantic details during denoising—leading to information loss and limiting both reconstruction fidelity and robustness. Overall, tensor-based defenses face a reconstruction–denoising conflict: strong low-rank suppression removes noise but sacrifices detail, whereas high-fidelity reconstruction risks recovering adversarial residues. These observations motivate our frequency-aware, progressive framework with resolution-dependent capacity control.

## 2.3 LAPLACIAN PYRAMID ARCHITECTURES

The Laplacian pyramid (LP), originally introduced for compact image coding (Burt & Adelson, 1987), decomposes an image into a low-frequency base and progressive band-pass residuals, each reconstructed by upsampling the coarser scale and adding the residual. Later studies highlighted that residuals should be treated as meaningful signals, not mere by-products, since explicitly modeling them improves stability and robustness to perturbations (Do & Vetterli, 2001). This idea has inspired architectures across vision: in segmentation, residual refinements sharpen boundaries without amplifying noise (Ghiasi & Fowlkes, 2016); in super-resolution and compressive sensing, networks like LapSRN and LAPRAN progressively predict residuals to stabilize training and avoid artifacts (Lai et al., 2017; 2018; Xu et al., 2018). Beyond pyramid decompositions, wavelet-based denoising techniques (Donoho & Johnstone, 1994) have demonstrated the effectiveness of constraining high-frequency components through shrinkage or thresholding. Such approaches support the broader intuition that residual channels, due to their larger representational freedom and broader bandwidth, are more prone to absorbing perturbations if left unconstrained. Introducing explicit regularization on residual components can therefore mitigate overfitting to noise while retaining essential structures. Furthermore, existing research Grabinski et al. (2022) has shown that the lack of control over high-frequency information during downsampling can amplify the model's sensitivity to high-frequency disturbances. These insights directly motivate our design, where progressive residual reconstruction is complemented with frequency-domain regularization to achieve a more balanced trade-off between fidelity and robustness.

## 3 PRELIMINARIES

### 3.1 PROBLEM SETUP AND THREAT MODEL

We consider an input image $x \in [0,1]^{H \times W \times C}$ with spatial size $H = W = 2^D$ and channel number $C$. A classifier $f(\cdot)$ maps the image to a label space. An adversary perturbs $x$ within a norm-bounded region to generate an adversarial example:

$$x^{\text{adv}} = x + \delta, \quad \|\delta\|_p \leq \varepsilon, \quad p \in \{2, \infty\}. \tag{1}$$

The goal of a purifier $P(\cdot)$ is to map the adversarial input back to a semantically faithful image $\hat{x} = P(x^{\text{adv}})$, such that $f(\hat{x})$ remains robust to adversarial perturbations while retaining high accuracy on clean data. We evaluate performance through both *clean accuracy* and *robust accuracy*.

### 3.2 TENSOR-NETWORK PRELIMINARIES

**Tensorization.** To leverage tensor-network models, an image (or its downsampled version at resolution $2^k \times 2^k$) is reshaped into a higher-order tensor. For example, the quantized representation maps a length-$N$ vector into a $d$-th order tensor of mode size 2:

$$N = 2^d, \quad x \in \mathbb{R}^N \mapsto \mathcal{X} \in \mathbb{R}^{2 \times 2 \times \cdots \times 2}. \tag{2}$$

This binary folding allows compact factorization by tensor networks.

**Tensor-Train (TT) Decomposition.** A $d$-th order tensor $\mathcal{X} \in \mathbb{R}^{n_1 \times n_2 \times \cdots \times n_d}$ can be represented in Tensor-Train (TT) format as a product of 3-way cores:

$$\mathcal{X}(i_1, i_2, \ldots, i_d) = A^1[i_1] \, A^2[i_2] \cdots A^d[i_d], \tag{3}$$

where each core $A^k \in \mathbb{R}^{r_{k-1} \times n_k \times r_k}$ and the TT-ranks $\{r_k\}$ control the expressivity of the representation with boundary conditions $r_0 = r_d = 1$.

**Quantized Tensor-Train (QTT).** In the quantized tensor-train format, each dimension $n_k$ of $\mathcal{X}$ is factorized into powers of two, enabling an exponential reduction in storage complexity (Loeschcke et al., 2024). Formally, for $n_k = 2^{m_k}$, the mode is folded into $m_k$ binary dimensions, and the resulting QTT representation achieves logarithmic scaling in the tensor order. This quantization step makes QTT especially suitable for representing high-resolution images compactly, while maintaining the low-rank bias critical for purification.

## 4 METHOD

### 4.1 PROBLEM FORMULATION

Among various AP approaches, tensor decomposition-based methods are particularly appealing because low-rank representations naturally capture the dominant semantics of images while suppressing redundant details (Bhattarai et al., 2023; Entezari & Papalexakis, 2022).

However, a fundamental difficulty arises in practice: adversarial purification typically has no access to clean references, and the model must operate directly on adversarially perturbed inputs. This creates an intrinsic conflict between data reconstruction and noise removal. Minimizing reconstruction error drives the tensor network to faithfully recover both clean content and adversarial perturbations, while enforcing strong denoising risks discarding important structural details needed for recognition (Lin et al., 2025). Existing methods that directly apply low-rank tensor approximations to input images often fall into this trade-off, yielding suboptimal purification performance.

Formally, given an input $x_D \in \mathbb{R}^{2^D \times 2^D}$, which may be clean or adversarial, and a target classifier $f$, the goal is to construct a tensor-network-based reconstruction operator $\mathcal{R}_\theta$ such that $\hat{x}_D = \mathcal{R}_\theta(x_D)$, $f(\hat{x}_D)$ is robust to adversarial perturbations. The central challenge is to design $\mathcal{R}_\theta$ that decouples the reconstruction of semantic content from the suppression of adversarial noise, thereby overcoming the inherent conflict embedded in existing direct-denoising approaches.

### 4.2 LAPLACIAN PYRAMID–BASED RECONSTRUCTION

Prior work (Lin et al., 2025) has established that, due to the central limit theorem, adversarial perturbations in downsampled images tend to converge toward Gaussian distributions. Consequently, even an $\ell_2$ penalty can effectively suppress perturbations at sufficiently low resolutions. However, if the reconstruction target is defined directly as multi-resolution representations of the original image, the perturbations will inevitably be reintroduced as the resolution increases, since high-frequency adversarial components are progressively recovered alongside the image details.

To address this intrinsic conflict, we design a reconstruction framework inspired by the Laplacian pyramid. Instead of reconstructing the full image at each resolution, we decompose the reconstruction task into disjoint frequency bands. At each stage, the model focuses only on recovering the clean component of the current frequency band. This design separates the structural content from the adversarial noise across scales and thereby mitigates the conflict between achieving high-fidelity reconstruction and removing perturbations when synthesizing the full-resolution image.

### 4.3 PROGRESSIVE RESIDUAL RECONSTRUCTION

Our method adopts a progressive residual reconstruction framework, as illustrated in Figure 1, which decouples coarse structural modeling from fine-scale detail refinement.

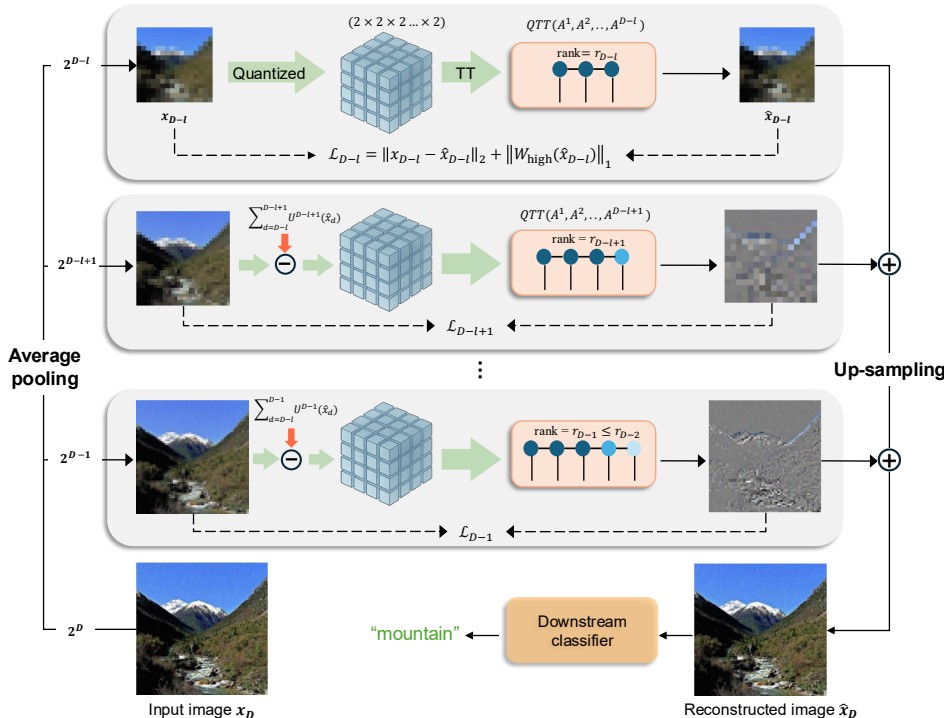

Figure 1: Illustration of tensor network purification.

Given an input image $x_D \in \mathbb{R}^{2^D \times 2^D}$, we construct a resolution pyramid $\{x_{D-l}\}_{i=0}^{l}$ through average pooling. At each level $k \in \{D-l, \ldots, D\}$, a tensor-network model is trained to predict an output $\hat{x}_k$ at resolution $2^k \times 2^k$.

**Training targets.** The training targets are defined separately for the base and refinement levels:

- **Base level** ($k = D - l$)**:** the target is simply the downsampled image,

$$\tau_{D-l} = x_{D-l}. \tag{4}$$

- **Refinement levels** ($k = D - l + 1, \ldots, D$)**:** the target is the Laplacian residual obtained by subtracting all coarser-level reconstructions (upsampled to the current resolution) from the original image,

$$\tau_k = x_k - \sum_{d=D-l}^{k-1} U^{k-d}(\hat{x}_d), \tag{5}$$

where $U(\cdot)$ denotes the $2\times$ upsampling operator.

**Per-level objective.** At level $k$, the tensor network produces $\hat{x}_k = \mathcal{T}_k(\theta_k)$. The stage-wise optimization objective at iteration $t$ is

$$\mathcal{L}_k = \left\| \tau_k - \hat{x}_k \right\|_2^2. \tag{6}$$

This residual design ensures that each level focuses only on the frequency band not yet captured by previous levels.

**Rank truncation for capacity control.** After each reconstruction stage, the QTT representation is upsampled to the next resolution via MPO and then truncated by TSVD to remain below a prescribed maximum rank. Crucially, we schedule this maximum rank to decrease as the resolution increases, i.e.,

$$r_{D-1} \le r_{D-2} \le \cdots \le r_{D-\ell}, \tag{7}$$

thereby progressively constraining the number of trainable parameters. This resolution-dependent rank scheduling gradually decreases the number of trainable parameters as the frequency content increases, thereby limiting the network's ability to reproduce high-frequency perturbations while preserving sufficient capacity to capture the global structures necessary for robust classification.

---

**Algorithm 1** Progressive Residual Reconstruction

---

**Input:** Input image $x_D$, pyramid levels $l$, pooling $D(\cdot)$, MPO upsampling $U(\cdot)$, steps $T_k$.
**Output:** Reconstructed image $\hat{x}_D$

---

**(1) Build multi-resolution pyramid**
**for** $k = D - l, D - l + 1, \ldots, D$ **do**
$\quad x_k \leftarrow D^k(x_D)$
**end for**
**(2) Base reconstruction**
$\tau_{D-l} \leftarrow x_{D-l}$
**for** $t = 1, \ldots, T_{D-l}$ **do**
$\quad \hat{x}_{D-l} \leftarrow \hat{x}_{D-l} - \beta \nabla_{\hat{x}_{D-l}} \mathcal{L}_{D-l}$
**end for**

**(3) Residual reconstruction**
**for** $k = D - l + 1, \ldots, D$ **do**
$\quad \tau_k \leftarrow x_k - \sum_{d=D-l}^{k-1} U^{k-d}(\hat{x}_d)$
$\quad$ **for** $t = 1, \ldots, T_k$ **do**
$\quad\quad \hat{x}_k \leftarrow \hat{x}_k - \beta \nabla_{\hat{x}_k} \mathcal{L}_k$
$\quad$ **end for**
**end for**
**(4) Final composition**
$\hat{x}_D \leftarrow \sum_{k=D-l}^{D} U^{D-k}(\hat{x}_k)$
**Return:** $\hat{x}_D$

---

**Final reconstruction.** During inference, the final reconstruction is obtained by summing the base reconstruction and all residual predictions upsampled to the finest resolution:

$$\hat{x}_D = \sum_{k=D-l}^{D} U^{D-k}(\hat{x}_k). \tag{8}$$

This design ensures that global structure is captured at coarse levels, while finer levels only supply the missing frequency bands, thereby improving stability and reducing the risk of reproducing adversarial noise.

### 4.4 WAVELET-BASED RESIDUAL REGULARIZATION

Recent studies have shown that adversarial perturbations often have strong effects on the high-frequency components of images. Pei et al. (2025) analyzed adversarial purification in the frequency domain and reported that perturbations are frequently entangled with fine-scale details. Similarly, Entezari & Papalexakis (2022) demonstrated that suppressing high-frequency content through tensor factorizations can improve robustness.

Motivated by these findings and supported by our own frequency-domain experiments (see Fig. 2 (a)), we introduce Wavelet-Based Residual Regularization (WRR) to explicitly control the high-frequency energy in the reconstruction process. For each reconstructed image, we apply a Haar wavelet transform and compute the $\ell_1$ norm of the horizontal, vertical, and diagonal sub-bands as a measure of high-frequency energy. This term is added to the total loss with a weight $\lambda$:

$$\mathcal{L}_k = \|\tau_k - \hat{x}_k\|_2^2 + \lambda \|W_{\text{high}}(y_k)\|_1, \tag{9}$$

where $W_{\text{high}}(\cdot)$ extracts the concatenated high-frequency subbands.

This regularization has two main benefits. First, it prevents residual channels from overfitting perturbations, which often exploit the flexibility of fine-scale components. Second, it stabilizes the progressive reconstruction process by ensuring that high-resolution stages only add necessary structural details instead of spurious noise. Importantly, WRR does not assume that all adversarial perturbations are purely high-frequency. Rather, it makes use of the fact that residual layers in pyramid-like architectures have greater freedom and are therefore more vulnerable to absorbing perturbations. By penalizing excessive high-frequency energy, our method achieves a better balance between semantic fidelity and robustness. The detailed algorithm is shown in Algorithm 1.

## 5 EXPERIMENTS

We evaluate our method on CIFAR-10, CIFAR-100, and ImageNet using pretrained ResNet and WideResNet classifiers, measuring both clean accuracy and robust accuracy as well as reconstruc-

tion quality metrics (PSNR, SSIM, NRMSE). Experiments are conducted under AutoAttack with $\ell_\infty$ and $\ell_2$ threat models, and comparisons are made against strong AT and AP baselines. In addition, we perform a set of ablation studies to examine the role of each design component: (i) reconstruction strategies (direct reconstruction, progressive upsampling, and our Laplacian pyramid–inspired residual reconstruction), (ii) resolution-dependent rank scheduling, and (iii) the proposed wavelet-based residual regularization by varying the coefficient $\lambda$. We also provide a frequency-domain analysis by tracking wavelet $\ell_1$ energy under different attack strengths. Together, these experiments highlight both the overall effectiveness of our method in mitigating adversarial perturbations and the contribution of each design choice to the trade-off faithful reconstruction and noise suppression.

## 5.1 EXPERIMENTAL SETUP

**Datesets and models** All our experiments are based on the CIFAR-10, CIFAR-100 (Krizhevsky et al., 2009), and ImageNet (Deng et al., 2009) datasets. For classification tasks, we utilize pre-trained ResNet (He et al., 2016) and WideResNet (Zagoruyko & Komodakis, 2016) models.

**Adversarial attacks methods** We evaluate our method with strong adaptive attacks. We use the commonly used AutoAttack $\ell_2$ and $\ell_\infty$ threat models (Croce & Hein, 2020), a widely used benchmark that integrates both white-box and black-box attacks.

## 5.2 COMPARISON WITH THE STATE-OF-THE-ART METHODS

**Robustness analysis against AutoAttack** Table 1 summarizes its performance under AutoAttack across CIFAR-10, CIFAR-100, and ImageNet with both $\ell_\infty$ and $\ell_2$ threat models. On CIFAR-10, our method achieves 73.04% robust accuracy under $\ell_\infty$ and 81.25% under $\ell_2$, surpassing prior defenses without additional data. On the more challenging CIFAR-100, our defense obtains 46.09% robust accuracy, outperforming comparable purification-based methods. On ImageNet, where scaling defenses is particularly difficult, our approach reaches 48.82% robust accuracy, significantly higher than previous baselines. These results demonstrate that the proposed design consistently improves robustness across datasets and threat models while maintaining competitive standard accuracy.

Table 1: Standard and robust accuracy (%) against AutoAttack under different threat models and datasets. "Extra" indicates using additional real data; $^\dagger$ marks additionally using *synthetic* images; $^*$ denotes using a robust classifier (Cui et al., 2024).

**CIFAR-10 ($\ell_\infty$, WideResNet-28-10, $\epsilon$=8/255)**

| Defense | Extra | Standard Acc. | Robust Acc. |
|---|---|---|---|
| Zhang et al. (2020) | ✓ | 85.70 | 59.90 |
| Bai et al. (2024) | ✓$^\dagger$ | 95.23 | 68.06 |
| Rebuffi et al. (2021) | ✗$^\dagger$ | 87.32 | 61.72 |
| Gowal et al. (2021) | ✗$^\dagger$ | 88.74 | 66.11 |
| Cui et al. (2024) | ✗$^\dagger$ | 92.16 | 67.37 |
| Nie et al. (2022) | ✗ | 89.02 | 70.64 |
| Lin et al. (2024) | ✗ | 90.62 | 72.84 |
| Lin et al. (2025)$^*$ | ✗ | 91.99 | 72.55 |
| **Ours** | ✗ | 82.61 | 50.97 |
| **Ours**$^*$ | ✗ | 90.03 | 73.04 |

**CIFAR-10 ($\ell_2$, WideResNet-28-10, $\epsilon$=0.5)**

| Defense | Extra | Standard Acc. | Robust Acc. |
|---|---|---|---|
| Augustin et al. (2020) | ✓ | 92.03 | 77.93 |
| Gowal et al. (2020) | ✓ | 94.74 | 80.56 |
| Rebuffi et al. (2021) | ✗$^\dagger$ | 91.79 | 78.33 |
| Ding et al. (2018) | ✗ | 88.02 | 67.77 |
| Nie et al. (2022) | ✗ | 90.89 | 78.58 |
| Lin et al. (2025)$^*$ | ✗ | 91.59 | 79.49 |
| **Ours** | ✗ | 82.61 | 64.06 |
| **Ours**$^*$ | ✗ | 90.03 | 81.25 |

**CIFAR-100 ($\ell_\infty$, WideResNet-28-10, $\epsilon$=8/255)**

| Defense | Extra | Standard Acc. | Robust Acc. |
|---|---|---|---|
| Hendrycks et al. (2019) | ✓ | 59.23 | 28.44 |
| Debenedetti et al. (2023) | ✓ | 70.76 | 35.20 |
| Cui et al. (2024) | ✗$^\dagger$ | 73.85 | 39.18 |
| Wang et al. (2023) | ✗$^\dagger$ | 75.11 | 42.67 |
| Pang et al. (2022) | ✗ | 63.66 | 31.10 |
| Jia et al. (2022) | ✗ | 67.31 | 31.91 |
| Cui et al. (2024) | ✗ | 65.73 | 32.52 |
| Lin et al. (2025) | ✗ | 62.30 | 44.44 |
| **Ours**$^*$ | ✗ | 66.79 | 46.09 |

**ImageNet ($\ell_\infty$, ResNet-50, $\epsilon$=4/255)**

| Defense | Extra | Standard Acc. | Robust Acc. |
|---|---|---|---|
| Engstrom et al. (2019) | ✗ | 62.07 | 31.06 |
| Wong et al. (2020) | ✗ | 55.62 | 26.95 |
| Salman et al. (2020) | ✗ | 64.02 | 37.88 |
| Bai et al. (2021) | ✗ | 67.38 | 35.51 |
| Nie et al. (2022) | ✗ | 67.79 | 40.90 |
| Chen & Lee (2024) | ✗ | 68.56 | 40.60 |
| Lin et al. (2025) | ✗ | 65.43 | 42.76 |
| **Ours** | ✗ | 67.18 | 48.82 |

**Frequency-domain analysis of adversarial perturbations** Figure 2 (a) presents a frequency-domain analysis of adversarial perturbations on ImageNet and CIFAR-10. We compute the wavelet-

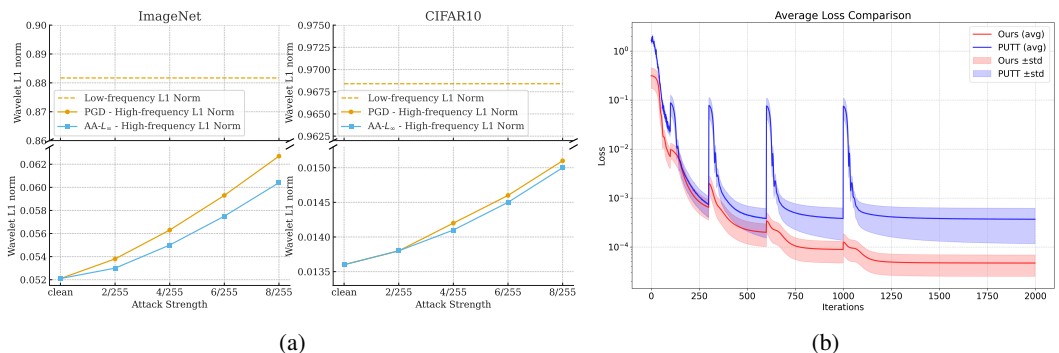

(a)                            (b)

Figure 2: (a) Effects of Different Attack Intensities on Different Frequency Bands of Images. (b) Average training loss comparison between the proposed method (Ours) and PUTT on Cifar10.

based energy of high- and low-frequency components using Haar filters under PGD and AutoAttack with varying attack strengths. The results show that the energy of the low-frequency components remains nearly unchanged, while the energy of the high-frequency components increases consistently as the attack strength grows. This indicates that adversarial perturbations primarily manifest in the high-frequency bands, providing strong motivation for our Wavelet-Based Residual Regularization. By explicitly constraining the high-frequency residuals during reconstruction, our method directly targets the spectral region where adversarial noise accumulates, thereby improving robustness without sacrificing the semantic content encoded in low-frequency components.

**Loss Convergence on CIFAR-10** Figure 2 (b) compares the average training loss curves of our Laplacian pyramid–based reconstruction with PUTT on CIFAR-10. Both methods follow a progressive paradigm, but they differ in how intermediate scales are handled: PUTT reconstructs the image directly from coarse to fine resolutions, while our approach reconstructs residuals at each stage using a Laplacian pyramid design. The results show that our method consistently achieves lower loss and converges more stably across iterations. In contrast, PUTT exhibits repeated loss spikes whenever the resolution is increased, reflecting difficulties in fitting the next finer scale. Our residual-based formulation mitigates this issue because each stage only learns the unexplained band, rather than re-estimating the entire image at the new resolution. This decomposition improves optimization conditioning, reduces redundancy across scales, and leads to smoother and faster convergence.

### 5.3 Ablation Experiment

**Analysis of results from different reconstruction strategies** To further illustrate the advantages of the Laplacian pyramid architecture for image reconstruction and denoising, we conduct an ablation study on ImageNet. Table 2 reports the reconstruction quality on both clean and adversarial images under different strategies, together with the classification performance of a ResNet-50 classifier on

Table 2: Comparison of reconstruction and classification performance under different reconstruction strategies on ImageNet (green: clean accuracy, red: robust accuracy).

| Reconstruction strategies | Resolution | QTT rank | PSNR | SSIM | NRMSE | Acc (%). |
|---|---|---|---|---|---|---|
| Direct reconstruction | 256 | 50 | 31.97
31.72 | 0.8722
0.8610 | 0.0547
0.0560 | 65.79
48.00 |
| Progressive reconstruction | 128→256 | 50-50 | 35.61
34.99 | 0.9211
0.9101 | 0.0387
0.0406 | 71.48
40.23 |
| Laplacian pyramid–based reconstruction | 128→256 | 50-50 | 36.87
36.20 | 0.9410
0.9313 | 0.0326
0.0344 | 72.85
36.52 |
| | | 50-40 | 35.09
34.58 | 0.9204
0.9099 | 0.0396
0.0412 | 72.07
43.16 |
| | | 50-30 | 33.36
32.98 | 0.8955
0.8848 | 0.0477
0.0492 | 67.96
48.39 |
| | | 50-20 | 31.68
31.41 | 0.8684
0.8567 | 0.570
0.0583 | 62.69
50.58 |
| | | 50-10 | 29.94
29.77 | 0.8360
0.8238 | 0.0681
0.0692 | 58.20
50.39 |

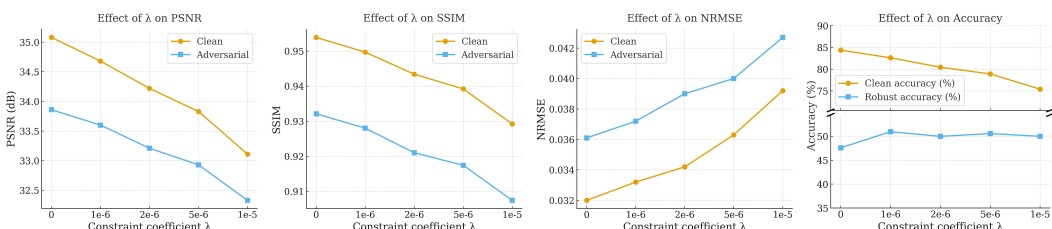

Figure 3: Effect of the constraint coefficient ($\lambda$) on model reconstruction and denoising performance.

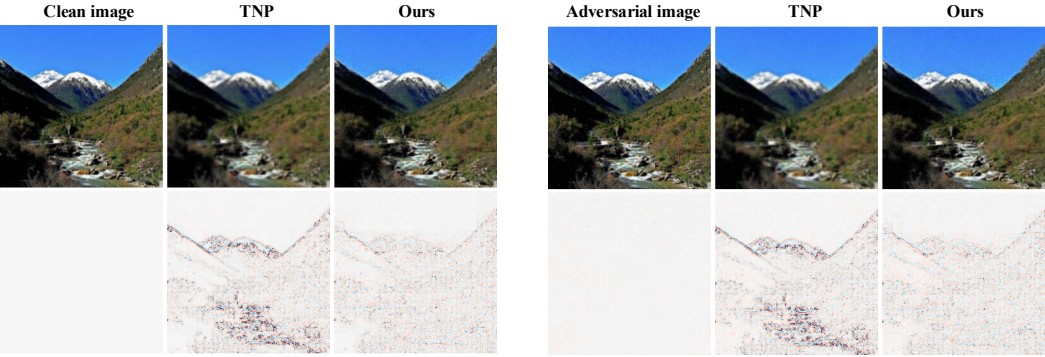

Figure 4: Illustration of reconstruction results under clean (left) and adversarial (right) inputs. The second row shows residuals with respect to the clean image.

the reconstructed images. To study the effect of reconstruction design, we consider three strategies — directly reconstructs the original image, progressively reconstructs from low to high resolution, and leverages the Laplacian pyramid for reconstruction.

From the results, direct reconstruction shows the lowest reconstruction quality and clean accuracy (65.79%), as tensor networks struggle to directly model high-resolution images. Progressive reconstruction improves both fidelity and clean accuracy, but repeated fitting across scales reduces robust accuracy. In contrast, Laplacian pyramid–based reconstruction achieves the best overall quality under the same parameter budget. However, unconstrained recovery of noisy high-frequency components leads to a sharp drop in robust accuracy (36.52%). By applying rank constraints to limit such components, robustness improves substantially while reconstruction quality and clean accuracy remain almost unchanged. These findings demonstrate that the Laplacian pyramid framework achieves a superior trade-off between reconstruction and effective noise suppression.

**Effectiveness of wavelet-based residual regularization** Figure 3 shows how varying $\lambda$ influences reconstruction and classification. Increasing $\lambda$ slightly reduces clean reconstruction fidelity, as residual details are suppressed, but at the same time consistently improves robust accuracy from 47% to about 50%. Beyond $\lambda = 1e-6$, robustness remains stable while clean accuracy gradually decreases, suggesting that an appropriate choice of $\lambda$ enables wavelet-based residual regularization to effectively suppress adversarial noise and achieve a favorable trade-off between fidelity and robustness.

**Visualization** Figure 4 compares reconstruction results of TNP and our method under both clean and adversarial inputs. Additionally, we highlight differences in the reconstruction by generating an error map between the reconstructed result and the clean image (second row). For clean inputs, our method produces reconstructions whose residuals are clearly weaker than those of TNP, indicating higher fidelity and fewer spurious artifacts. For adversarial inputs, when both reconstructions are compared against the clean target, our method again yields smaller residuals, suggesting that it better suppresses perturbation-related deviations while preserving the underlying semantic structure.

## 6 CONCLUSION

We introduced a residual-guided tensor network framework for adversarial purification that combines progressive Laplacian pyramid reconstruction, resolution-dependent rank scheduling, and

wavelet-based residual regularization. This design alleviates the conflict between reconstruction fidelity and denoising, enabling a principled balance between clean accuracy and adversarial robustness. Extensive experiments on CIFAR-10, CIFAR-100, and ImageNet demonstrate that our method effectively suppresses adversarial perturbations while preserving semantic fidelity, validating the potential of tensor-network–based purification for reliable defense.

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

## A    APPENDIX

### LLM USAGE DISCLOSURE

We made use of large language models (LLMs) as writing assistants throughout the preparation of this paper. Specifically, LLMs were employed not only for grammar correction and paraphrasing, but also for helping to restructure certain sections, refine the clarity of technical descriptions, and improve overall readability. All scientific content, research ideas, experiments, and conclusions were conceived, developed, and verified by the authors. The role of LLMs was supportive in enhancing the presentation and accessibility of the text, while the responsibility for the accuracy and integrity of the content rests entirely with the authors.

## B    IMPLEMENTATION DETAILS OF THE EXPERIMENT

### B.1    ADVERSARIAL ATTACKS

We evaluated the defense performance of our method on ImageNet and CIFAR-10 under AutoAttack. The compared methods include several state-of-the-art AP/AT approaches listed in the RobustBench benchmark. To ensure fair comparison, we follow the standard threat-model settings commonly used in adversarial defense research. Specifically, for CIFAR-10, we set the AutoAttack $l_\infty$ and $l_2$ perturbation budgets to $\epsilon = 8/255$ and $\epsilon = 0.5$, respectively. For CIFAR-100, we use AutoAttack $l_\infty$ with $\epsilon = 8/255$. For ImageNet, we follow the standard setup of AutoAttack $l_\infty$ with $\epsilon = 4/255$.

It is important to note that applying PGD+EOT (Madry et al., 2017; Athalye et al., 2018) directly to attack the entire pipeline (purifier + classifier) is not feasible. Unlike neural-network-based models, tensor networks do not contain trainable parameters. Learning a representation with a tensor network is an optimization process: only after an input image is provided does the tensor network perform decomposition and reconstruction to generate its low-rank representation. Furthermore, our purification pipeline contains several non-differentiable steps—including downsampling, tensor-network-based reconstruction, and upsampling— which makes the overall mapping non-differentiable. As a consequence, the entire process is non-differentiable, making full gradient attacks unable to obtain meaningful gradients to effectively attack our method.

Therefore, following prior adaptive attack practices (Lin et al., 2024; Yang et al., 2019), we adopt BPDA as the adaptive attack strategy. During the backward pass, the purification module is approximated as an identity mapping. Consistent with the settings in (Lee & Kim, 2023), we use 200 PGD iterations and 20 EOT samples during the attack. We compared several AP/AT methods, the experimental results are reported below.

### B.2    OUR METHOD

In our purification pipeline, the QTT representation requires the spatial dimensions to be powers of two. Therefore, the original images from CIFAR-10 ($32 \times 32$) and ImageNet ($224 \times 224$) are first upsampled to $64 \times 64$ and $256 \times 256$, respectively. This adjustment not only satisfies the QTT requirement but also provides sufficient room for multi-scale decomposition. In particular,

directly downsampling a $32 \times 32$ CIFAR-10 image to very low resolutions (e.g., 4 or 8) would severely damage semantic content and hinder reconstruction; thus, upsampling to $64 \times 64$ ensures a meaningful frequency hierarchy.

After resizing, the images are decomposed into multiple scales following a Laplacian pyramid–style architecture. For ImageNet, the $256 \times 256$ image is downsampled progressively to resolutions [16, 32, 64, 128]. For CIFAR-10, the $64 \times 64$ image is downsampled to [8, 16, 32]. These multi-resolution components provide frequency-separated targets that support stable coarse-to-fine reconstruction.

At the coarsest level, the reconstruction target is the low-resolution version of the original image, which primarily contains low-frequency structural information. For all higher levels, the reconstruction target is defined as the residuals between adjacent resolutions, corresponding to high-frequency components such as edges and textures. Consequently, higher TT-ranks are required at low resolutions to preserve global structures, while lower ranks are sufficient at higher resolutions where the reconstruction focuses on local high-frequency details. This rank scheduling strategy enables high-quality reconstruction while effectively suppressing adversarial perturbations. Following this principle, we set the rank at the coarsest level equal to its spatial resolution, and progressively reduce the ranks at finer levels to values below their corresponding resolutions. Specifically, for ImageNet, we use the multi-scale resolutions [256,128,64,32,16], with the corresponding QTT ranks set to [29,30,31,32,16]. For CIFAR-10 and CIFAR-100, the downsampled resolutions are [64,32,16,8], and the associated QTT ranks are [4,10,16,8].

## C  ADDITIONAL RESULTS

### C.1  RESULTS ANALYSIS ON DIFFERENT ATTACKS

we additionally evaluated our method using **Diff-PGD** (Xue et al., 2023). The attack was configured according to the settings as follow:

diffusion model accelerator: 50

reverse steps in SDEdit: 2

$l_\infty$ PGD budget: 8/255

PGD iterations: 2

The resulting performance is shown in Table 3. The results show that, under settings where the clean accuracy is matched, our method achieves approximately 6% higher robust accuracy compared to the tensor-network–based purification baseline (TNP) (Lin et al., 2025).

Table 4 shows the comparison of robust accuracy against **PGD+EOT** with $l_\infty(\epsilon = 8/255)$ on CIFAR-10 with WideResNet-28-10. Consistent with the settings in (Lee & Kim, 2023), we use 200 PGD iterations and 20 EOT samples during the attack. The experimental results show that compared with the AP method, when achieving the same clean accuracy rate, the robust accuracy rate of our method is 11.89% higher than that of the second place. For all AT/AP methods, our approach achieves the optimal trade-off between clean accuracy and robust accuracy.

Table 3: Combined reconstruction and robustness performance under Diff-PGD attack.

| Method | PSNR | SSIM | NRMSE | Clean Acc. | Robust Acc. |
|--------|------|------|-------|-----------|-------------|
| TNP | 31.10 / 30.32 | 0.9040 / 0.8609 | 0.0574 / 0.0618 | 65.20 | 34.00 |
| Ours | 32.99 / 31.82 | 0.8975 / 0.8570 | 0.0486 / 0.0540 | 64.00 | 40.00 |

### C.2  ABLATION ON WAVELET SUB-BAND REGULARIZATION

We evaluate how penalizing different wavelet sub-bands (HH, HL, LH) combinations influences reconstruction and robustness. By applying Haar decomposition to the reconstructed image and minimizing the $l_1$-norm of selected high-frequency components, we tested the effect of constraining

Table 4: Standard accuracy and robust accuracy against PGD-EOT attack $l_\infty$ ($\epsilon = 8/255$) on CIFAR-10 with WideResNet-28-10 classifier.

| Type | Method | Clean Acc. | Robust Acc. |
|---|---|---|---|
| AT | Gowal et al. (2020) | 87.51 | 66.01 |
| | Gowal et al. (2021) | 88.71 | 65.93 |
| | Pang et al. (2022) | 88.62 | 64.95 |
| AP | Yoon et al. (2021) | 85.66 | 33.48 |
| | Nie et al. (2022) | 89.02 | 46.84 |
| | Lee & Kim (2023) | 91.99 | 55.82 |
| | Ours | 90.03 | 67.71 |

different high-frequency sub-bands on suppressing adversarial noise. The experimental results are shown in the Table 5. Based on the experimental results, Several observations emerge:

**No single subband dominates.** Applying penalties to individual subbands yields comparable reconstruction quality and robustness, indicating adversarial perturbations are not confined to a single frequency direction.

**Pairwise penalties prove unstable.** Combining two subbands yields inconsistent results, occasionally improving robustness accuracy marginally. This suggests selective regularization of high-frequency spectral components leads to imbalanced perturbation suppression.

**Simultaneously penalizing all high-frequency bands proves most stable and effective.** Applying WRR to the full band set (LH+HL+HH) achieves the optimal tradeoff between clean accuracy and robust accuracy across all variants. This validates our design choice: adversarial perturbations involve multi-directional components, and penalizing the concatenated high-frequency representation provides the most reliable control without overfitting to specific sub-bands.

Table 5: Combined reconstruction and robustness performance across different wavelet bands.

| Method | PSNR | SSIM | NRMSE | Clean Acc. | Robust Acc. |
|---|---|---|---|---|---|
| LH | 33.29 / 32.93 | 0.8991 / 0.8877 | 0.0484 / 0.0499 | 68.21 | 47.66 |
| HL | 33.24 / 32.89 | 0.8984 / 0.8871 | 0.0486 / 0.0501 | 68.99 | 47.10 |
| HH | 33.80 / 33.37 | 0.9057 / 0.8944 | 0.0465 / 0.0481 | 68.79 | 47.85 |
| LH + HL | 32.65 / 33.35 | 0.8892 / 0.8777 | 0.0512 / 0.0526 | 68.51 | 47.80 |
| LH + HH | 32.98 / 32.74 | 0.8946 / 0.8856 | 0.0497 / 0.0508 | 68.01 | 48.41 |
| HL + HH | 32.94 / 32.63 | 0.8941 / 0.8827 | 0.0498 / 0.0512 | 67.79 | 48.58 |
| LH + HH + HL | 33.57 / 33.19 | 0.9032 / 0.8919 | 0.0472 / 0.0487 | 68.16 | 49.21 |

C.3 EFFECT OF DOWNSAMPLING DEPTH AND RANK CONSTRAINTS

To better understand how the downsampling depth and the choice of QTT ranks influence reconstruction quality and adversarial purification performance, we conduct an ablation study using different multi-scale settings. The results are summarized in Table 6.

We vary the lowest-resolution level from $128 \rightarrow 256$ down to $8 \rightarrow 256$, and evaluate both fixed-rank configurations (e.g., 50, 50, . . . ) and our resolution-aware rank schedule (e.g., 16, 32, 31, 30, 29). Several observations emerge:

**Increasing the downsampling depth improves reconstruct quality but slightly harms robustness.** While maintaining the rank unchanged, as the number of layers deepened (from 2 to 5), the PSNR increased by 1.66, and the clean accuracy rate also rose by 1.56% accordingly. This indicates that deeper downsampling is conducive to improving the reconstruction quality of images by tensor networks. However, the robustness accuracy decreased by 6.84%, indicating that deeper,

unconstrained reconstructions are also easier to recover adversarial perturbations. These results emphasize that although progressive approaches are beneficial for reconstruction, appropriate capacity control (e.g., rank scheduling) is necessary to prevent the restoration of adversarial perturbations.

**Our adaptive rank schedule provides the best robustness–quality trade-off.** The configuration 16→256 with ranks [16,32,31,30,29] achieves the highest robust accuracy (49.21%) while maintaining competitive reconstruction metrics. This confirms that our rank selection strategy is effective: higher ranks help capture global low-frequency structures at coarse levels, while lower ranks effectively suppress perturbation-related high-frequency components at finer levels.

**Conclusion.** Both the depth of the decomposition and the rank constraints play crucial roles. Our rank-scheduled Laplacian pyramid finds a balanced middle ground, yielding strong robustness while maintaining good perceptual quality.

Table 6: Comparison of reconstruction and classification performance under different resolutions.

| Resolution | Rank | PSNR | SSIM | NRMSE | Clean Acc. | Robust Acc. |
|---|---|---|---|---|---|---|
| 128→256 | 50,50 | 36.87 / 36.20 | 0.9419 / 0.9313 | 0.0326 / 0.0344 | 72.85 | 36.52 |
| 64→256 | 50,50,50 | 37.99 / 37.34 | 0.9558 / 0.9473 | 0.0281 / 0.0298 | 73.00 | 31.64 |
| 32→256 | 50,50,50,50 | 38.10 / 37.40 | 0.9560 / 0.9477 | 0.0280 / 0.0298 | 73.85 | 30.83 |
| 16→ 256 | 50,50,50,50,50 | 38.44 / 37.58 | 0.9572 / 0.9483 | 0.0277 / 0.0297 | 74.41 | 29.68 |
| 8→ 256 | 50,50,50,50,50,50 | 38.45 / 37.58 | 0.9574 / 0.9484 | 0.0277 / 0.0295 | 74.44 | 29.66 |
| 16→ 256 | 16,32,31,30,29 | 33.57 / 33.19 | 0.9032 / 0.8919 | 0.0472 / 0.0487 | 68.16 | 49.21 |

C.4  MORE DETAILED EXPERIMENTS ON DIFFERENT RECONSTRUCTION STRATEGIES

As shown in Table 7, we provide a detailed comparison of different reconstruction strategies under varying downsampling depths. Two consistent findings emerge:

**Deeper reconstruction improves image fidelity but harms robustness.** For both direct and progressive reconstruction, increasing the number of reconstruction levels (e.g., from $128 \rightarrow 256$ to $16 \rightarrow 256$) consistently boosts PSNR/SSIM and reduces NRMSE. However, the adversarial accuracy drops ($40.23\% \rightarrow 38.99\%$), indicating that deeper reconstructions tend to overfit and restore adversarial perturbations along with image details. This shows that high-capacity, unconstrained reconstruction favors reconstruction fidelity but degrades purification effectiveness.

**Laplacian pyramid-based reconstruction provides the best trade-off.** The Laplacian Pyramid (LP) reconstructs different frequency bands of the image respectively, thus supporting hierarchical scheduling at each level. This enables us to limit the characterization of high-resolution subbands. As a result, LP-based reconstruction achieves the optimal balance between reconstruction quality and robustness. For instance, the $16 \rightarrow 256$ LP model with rank scheduling yields a higher robust accuracy (49.21%) than all progressive settings, while maintaining competitive reconstruction metrics (PSNR 33.57 / 33.19).

Overall, the experiments confirm that although deeper decompositions benefit reconstruction, rank constraints are essential to prevent restoring adversarial perturbations. The Laplacian pyramid framework, combined with rank scheduling, provides the most effective balance in practice.

# D  COMPUTATIONAL COMPLECITY

We analyzes in detail the computational complexity of each component in our implementation pipeline, including: (i) multi-scale image operations (upsampling and pooling), (ii) QTT decomposition and reconstruction, (iii) MPO-based prolongation, (iv) TT-SVD rank truncation, (v) residual update, and (vi) final multi-scale synthesis.

Let the spatial resolution at level $k$ be $H_k = 2^k$ with $C$ channels, and denote the maximum TT-rank by $r_k \leq r_{\max}$. The computational cost at level $k$ becomes

$$\mathcal{C}_k = I_k \times [\mathcal{O}(2^{2k}C) + \mathcal{O}(kr_k^2 2^{2k})] + \mathcal{O}(kr_k^3),$$

Table 7: Comparison of reconstruction and classification performance under different reconstruction strategies (green: clean accuracy, red: robust accuracy).

| Reconstruction strategies | Resolution | QTT rank | PSNR | SSIM | NRMSE | Acc (%). |
|---|---|---|---|---|---|---|
| Direct reconstruction | 256 | 50 | 31.97 / 31.72 | 0.8722 / 0.8610 | 0.0547 / 0.0560 | 65.79 / 48.00 |
| Progressive reconstruction | 128→256 | 50-50 | 35.61 / 34.99 | 0.9211 / 0.9101 | 0.0387 / 0.0406 | 71.48 / 40.23 |
| | 64→256 | 50-50-50 | 35.58 / 34.97 | 0.9211 / 0.9100 | 0.0388 / 0.0406 | 71.60 / 39.90 |
| | 16→256 | 50-50-50-50-50 | 36.10 / 35.32 | 0.9240 / 0.9147 | 0.0367 / 0.0400 | 71.67 / 38.99 |
| Laplacian pyramid–based reconstruction | 128→256 | 50-50 | 36.87 / 36.20 | 0.9410 / 0.9313 | 0.0326 / 0.0344 | 72.85 / 36.52 |
| | | 50-30 | 33.36 / 32.98 | 0.8955 / 0.8848 | 0.0477 / 0.0492 | 67.96 / 48.39 |
| | 64→256 | 50-50-50 | 37.99 / 37.34 | 0.9558 / 0.9473 | 0.0281 / 0.0298 | 73.00 / 31.64 |
| | 16→256 | 50-50-50-50-50 | 38.44 / 37.58 | 0.9572 / 0.9484 | 0.0277 / 0.0295 | 74.44 / 29.66 |
| | 16→256 | 16-32-31-30-29 | 33.57 / 33.19 | 0.9032 / 0.8919 | 0.0472 / 0.0487 | 68.16 / 49.21 |

where $I_s$ is the number of training iterations and the three terms correspond respectively to 2D image operations, QTT reconstruction, and MPO + TT-SVD rank truncation.

Since $H_k$ increases exponentially with $k$, the total complexity is ultimately dominated by the highest resolution, $H_{\max} = 2^D$. The total cost can therefore be approximated as

$$\mathcal{C}_{\text{forward}} = I_k \times [\mathcal{O}(H_{\max}^2 C) + \mathcal{O}(r_{\max}^2 H_{\max}^2 \log H_{\max})].$$

The first term reflects the cost of multi-scale 2D image processing, while the second term arises from QTT-based reconstruction.

## E  INFERENCE COST

we evaluated the efficiency of our method by measuring both inference time and the amount of computation involved during inference. We compared our approach with a representative diffusion-based purification method (DiffPure) (Nie et al., 2022) and a tensor-network–based purification baseline (TNP) (Lin et al., 2025). All measurements were conducted on a single CIFAR-10 image using an NVIDIA RTX A6000 GPU.

**Inference time.** As shown in Table 8, the results show that our approach is faster than DiffPure, and it also avoids the high computational cost of adversarial training. Compared with TNP, our method is slightly slower, mainly due to the additional optimization introduced by the wavelet-based regularization in our reconstruction process. This step provides the robustness benefits discussed in the paper, but also adds some overhead.

**Inference parameters.** To further quantify efficiency, we also compared the number of parameters involved in the inference process for the three methods. As shown in Table 9, diffusion-based purification requires orders of magnitude more parameters due to the heavy denoising backbone. In contrast, both tensor-network–based methods (TNP and ours) require significantly fewer parameters, with our method using the fewest among all three. This demonstrates that our approach maintains the lightweight nature of tensor-network–based purification while providing improved robustness.

These results prove that our method achieves competitive robustness while remaining computationally efficient compared to diffusion-based purification and comparable tensor-network baselines.

Table 8: Inference time of different methods for a single Cifar10 image on an NVIDIA RTX A6000 GPU.

| Methods | DiffPure | TNP | Ours |
|---------|----------|-----|------|
| Time (s) | 6.86 | 1.25 | 1.75 |

Table 9: Number of inference time parameters of different methods for a single Cifar10 image on an NVIDIA RTX A6000 GPU.

| Methods | DiffPure | TNP | Ours |
|---------|----------|-----|------|
| Trainable parameters (s) | 143,111,773 | 34369 | 13,756 |

# F  VISUALIZATION

## F.1  RECONSTRUCTION DIFFERENCE VISUALIZATION OF PUTT AND OURS

we conducted a visual comparison between the coarse-to-fine tensor-network reconstruction (PUTT) and our LP+WRR framework. The results are shown in Figure 5 in the supplementary material. For each method, we present three images: the clean input, the reconstruction of the adversarial input, and the pixel-wise difference between the two.

We observe a clear qualitative distinction between the two approaches. In PUTT, adversarial perturbations remain visible across both smooth and textured regions, and the difference map exhibits noise spread throughout the entire image. This indicates that the coarse-to-fine TT reconstruction tends to retain adversarial noise at multiple scales, consistent with the high flexibility of TT cores in fine-resolution stages.

In contrast, our LP+WRR method suppresses perturbations more effectively. The difference map shows that the remaining variations are concentrated mainly around edges and high-frequency structures, while large smooth regions are largely free of residual noise. This behavior aligns with our design: the wavelet-based regularization penalizes high-frequency subbands and prevents the reconstruction process from absorbing perturbations into the residual components.

These visual results provide supporting evidence that, although inspired by coarse-to-fine TT, our method incorporates additional mechanisms that mitigate the tendency of TT-based reconstruction to recover adversarial perturbations.

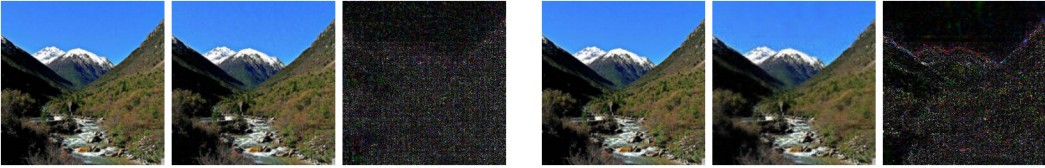

Figure 5: Visual comparison of reconstruction and denoising results between PUTT and our method.

## F.2  VISUALIZATION OF PROGRESSIVE RESIDUAL RECONSTRUCTION

Figure 6 to 8 shows the visualization results of progressive residual reconstruction based on clean inputs and adversarial inputs. The first row shows the clean (or adversarial) image and its downsampled versions at different resolutions. The second row illustrates the reconstruction targets at each scale (lowest-resolution base and residuals between consecutive levels). The third row presents the cumulative reconstruction results up to each scale, and the fourth row shows the error maps between reconstructed images and the clean input. Rows five to eight repeat the same process for adversarial inputs. These visualizations highlight how our method incrementally refines structures while constraining residuals, and how errors remain consistently small compared to the clean target.

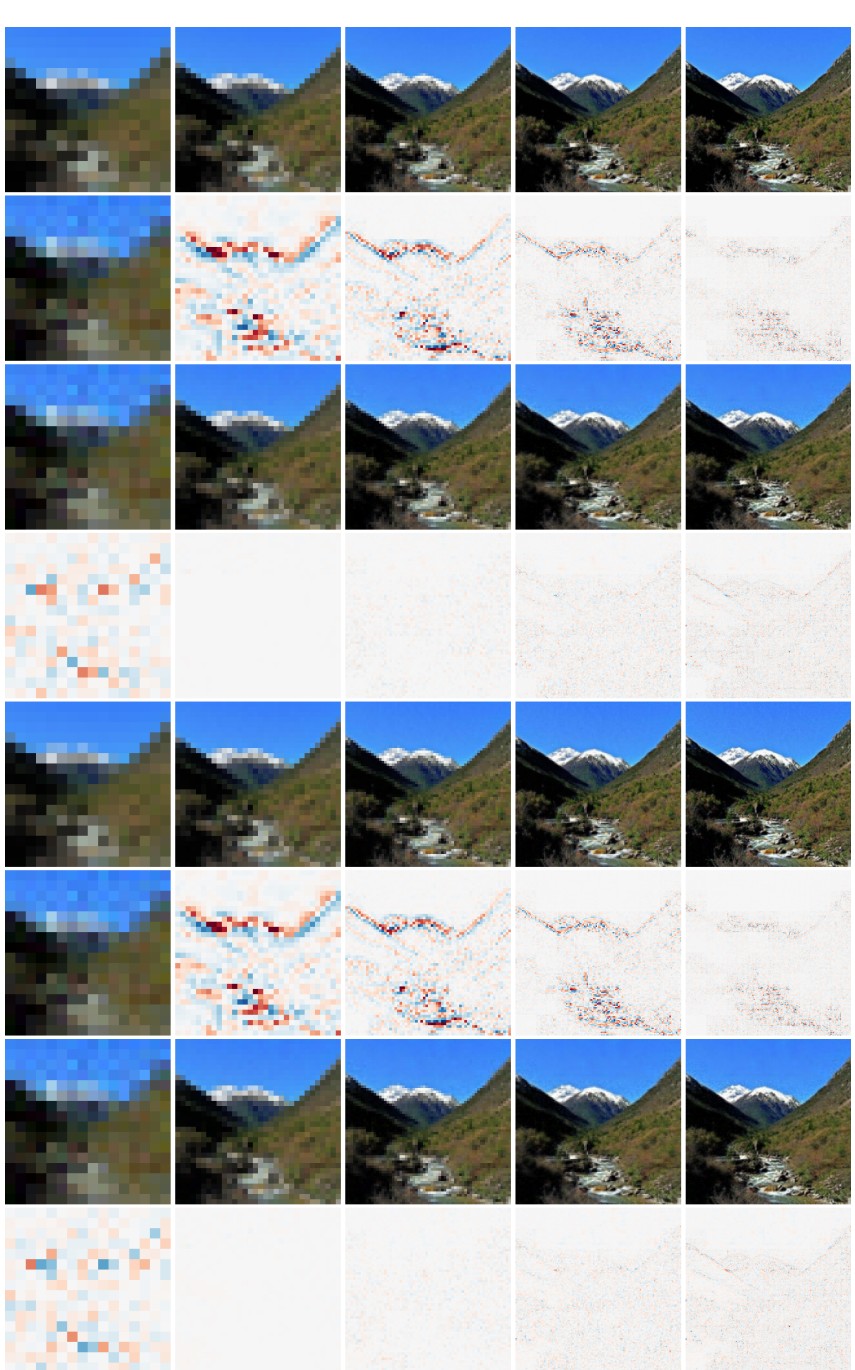

Figure 6: Visualization of Progressive Residual Reconstruction.

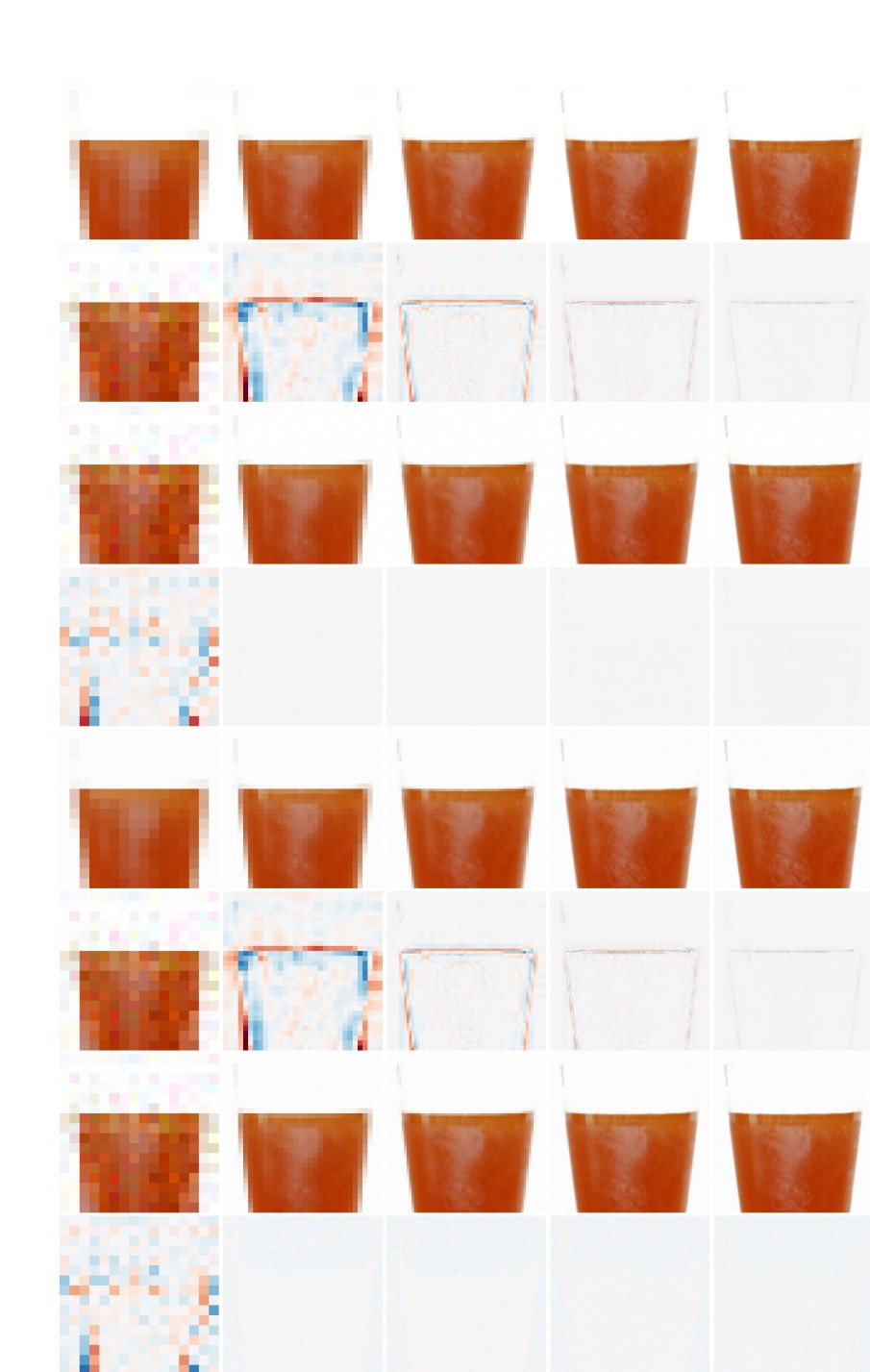

Figure 7: Visualization of Progressive Residual Reconstruction.

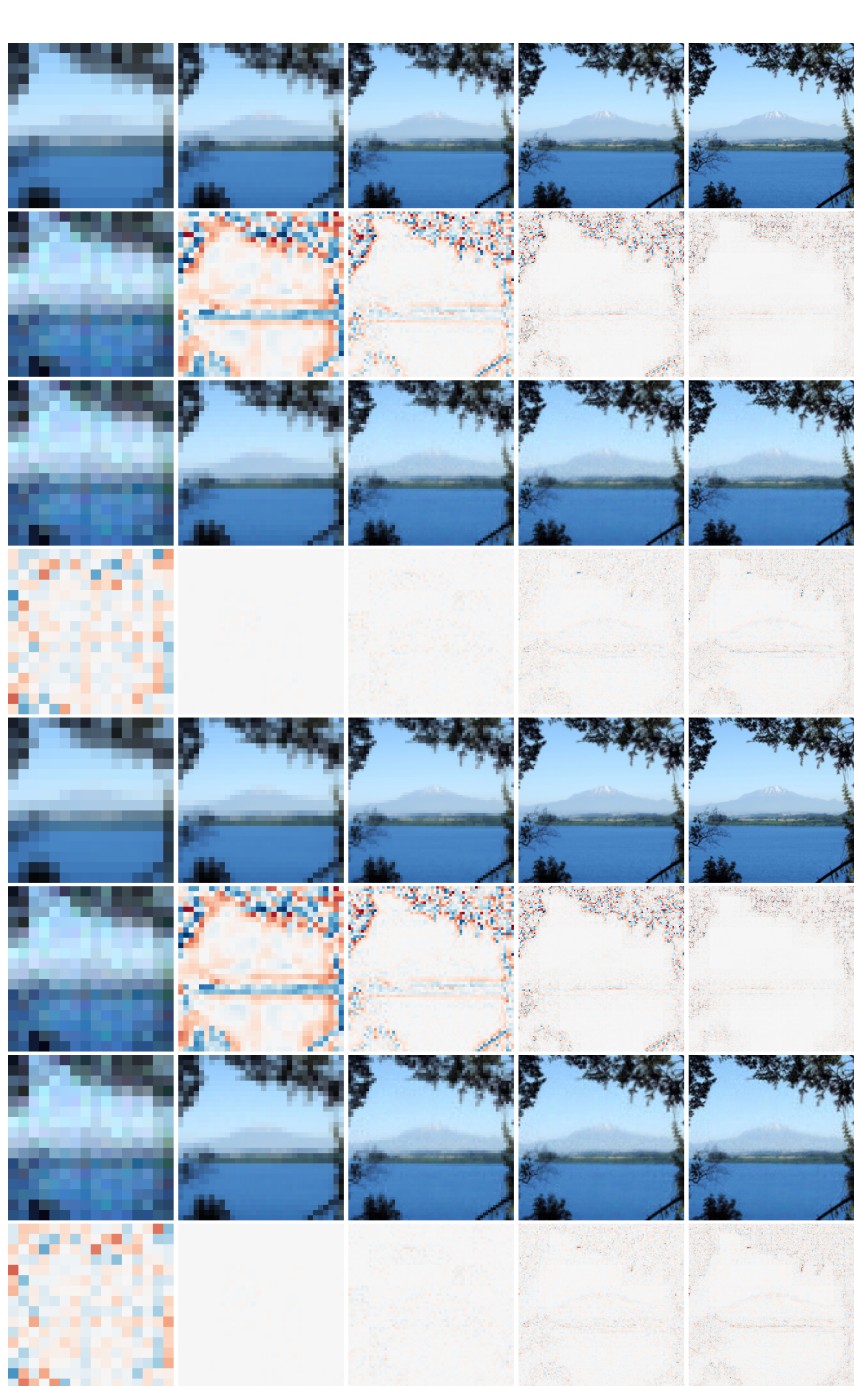

Figure 8: Visualization of Progressive Residual Reconstruction.

