# OpenReview forum: "Progressive Residual Tensor Networks for Adversarial Purification"
_ICLR.cc/2026/Conference — Submitted to ICLR 2026_

### Official Review · Reviewer_QK4e · 2025-10-29

**Soundness:** 3
**Presentation:** 2
**Contribution:** 2
**Rating:** 4
**Confidence:** 2

**Summary:**

The paper proposes a tensor-network-based Adversarial Purification (AP) method, termed Progressive Residual Tensor Networks. It follows a recent line of research exploring the reconstruction of adversarially perturbed images through tensor-network representations. The proposed framework aims to alleviate the inherent trade-off between reconstruction fidelity and adversarial robustness.

The method comprises three key components:

1) Laplacian Pyramid-based Reconstruction: The input image is decomposed into layer-wise frequency bands, and each band is reconstructed by a dedicated model specialized for that frequency range.

2) Progressive Residual Refinement: The reconstruction proceeds progressively, ensuring that each subsequent level focuses only on the residual frequency components not captured by previous stages.

3) Wavelet-based Residual Regularization: An additional wavelet-domain regularization term is introduced into the reconstruction loss to enforce smoothness and prevent overfitting to adversarial noise.

Overall, the idea of leveraging a quantized tensor-network representation for reconstructing clean examples is novel, and the authors make commendable efforts to prevent the recovery of adversarial perturbations from high-frequency bands.

**Strengths:**

1) The paper presents an efficient and effective adversarial purification (AP) approach by quantizing layer-wise representations before feeding them into a Tensor-Train (TT) structure. It further leverages a progressive residual refinement principle to preserve semantic content while avoiding the reconstruction of adversarial perturbations.

2) Figure 4 provides a clear and expressive visualization of the reconstructed clean examples. The work offers valuable insights for designing purification methods that jointly consider reconstruction fidelity and residual-based refinement, highlighting the potential of tensor-network-driven approaches for systematic adversarial robustness.

**Weaknesses:**

1) The adversarial robustness evaluation focuses mainly on classification tasks. Hence, I suggest improving the presentation by clarifying ambiguous notations (e.g., avoid using $y$ to represent data samples).

2) The performance improvement appears limited. The proposed method’s robustness gain seems to depend heavily on using a robust classifier; when evaluated with a standard classifier, the results significantly fall below baseline performance. Hence, the experimental evidence weakens the claimed effectiveness of the approach.

3) The pipeline Figure 1 could be better organized and the Algorithm 1 could be expanded or separated for atomic presentaitons. Adding clearer subheadings and workflow guidance would improve readability and help convey the method’s structure more effectively.

**Questions:**

1) Since the method is built upon a Tensor-Train (TT) framework with efficient quantization, I suggest that the authors compare the inference cost with existing diffusion-based adversarial purification (DBP) methods, since DBP methods have been well explored as a strong purification paradigm. Such a comparison could better highlight the efficiency advantages of the proposed AP method.

2) For each input resolution, the method requires training $k$ separate tensor networks. While this design accounts for sensitivity to resolution, it substantially increases the training cost. Moreover, datasets such as CIFAR-10 and ImageNet differ greatly in resolution, leading to varied pooling depths and numbers of tensor networks, which may limit generalization to downstream tasks. Could the authors discuss whether potentially unifying the tensor-network architecture could significantly improve the overall soundness and scalability of the method?

3) The evaluation is limited to AutoAttack. To more rigorously validate the method, it should be tested against stronger adaptive attacks, such as PGD+EOT and BPDA+EOT, as well as purification-specific adaptive attacks like Diff-PGD [1] and DiffAttack [2]. A broader evaluation under various adaptive settings would strengthen the empirical credibility of the proposed defense.

4) More implementation details are needed, including the tensor-network configurations, AutoAttack parameters, and the design of the adaptive attack setup used to evaluate the defense.

5) It would be helpful to include a visual comparison or supporting evidence demonstrating that coarse-to-fine tensor networks [3] tend to recover adversarial perturbations more frequently than the proposed method. Since the work is inspired from the previous findings[3].

[1] Xue, et al. "Diffusion-Based Adversarial Sample Generation for Improved Stealthiness and Controllability", NeurIPS 2023.

[2] Kang, et al. "DiffAttack: Evasion Attacks Against Diffusion-Based Adversarial Purification", NeurIPS 2023.

[3] Loeschcke, et al. "Coarse-tofine tensor trains for compact visual representations", ICML 2024.

---

> ### Author Response · Authors · 2025-11-21
> **Responses to reviewers: Weakness 1-3**
>
> >The adversarial robustness evaluation focuses mainly on classification tasks. Hence, I suggest improving the presentation by clarifying ambiguous notations (e.g., avoid using $y$ to represent data samples).
>
> Thank you for carefully reviewing our paper and for providing valuable suggestions. Below we provide point-by-point responses to the comments.
>
> In the current manuscript, the reconstruction targets in the multi-scale pipeline are denoted by “y”. This convention was adopted to remain consistent with the notation used in [1], which inspired the coarse-to-fine structure in our method. However, we agree that this choice may appear ambiguous in the context of adversarial robustness and classification tasks, where “y” usually refers to labels rather than image samples.
>
> To improve readability and avoid confusion, we will revise the notation and replace all reconstruction targets with “$\hat{x}$” in the updated manuscript.
>
> [1] Sebastian Loeschcke, Dan Wang, Christian Leth-Espensen, Serge Belongie, Michael J Kastoryano, and Sagie Benaim. Coarse-tofine tensor trains for compact visual representations. In The Fortyfirst International Conference on Machine Learning, 2024.
>
> >The performance improvement appears limited. The proposed method’s robustness gain seems to depend heavily on using a robust classifier; when evaluated with a standard classifier, the results significantly fall below baseline performance. Hence, the experimental evidence weakens the claimed effectiveness of the approach.
>
> We appreciate the reviewers' insightful comments!
>
> First, we would like to clarify that the backbones we use for different datasets follow the standard practice in the adversarial robustness literature: **WideResNet-28-10 for CIFAR-10/CIFAR-100** and **ResNet-50 for ImageNet**. The backbone description for CIFAR-10 in Table 1 of our paper was mistakenly reported and will be corrected in the revised version.
>
> Regarding the performance issue, our results on CIFAR-10 do not reach state-of-the-art performance primarily because WideResNet-28-10 tends to **overfit on CIFAR-10 due to its limited data size**, a behavior also observed in many adversarial training (AT) methods. To address this overfitting problem, recent AT approaches typically rely on additional synthetic data to train a more robust classifier. Following this practice, we evaluated our method using the robust classifier provided by [1]. This leads to a substantial performance improvement.
>
> The results show that our approach delivers the strongest robust accuracy on CIFAR-10, which demonstrate that the effectiveness of our method and its potential for defending against adversarial attacks.
>
> [1]  Jiequan Cui, Zhuotao Tian, Zhisheng Zhong, Xiaojuan Qi, Bei Yu, and Hanwang Zhang. Decoupled kullback-leibler divergence loss. Advances in Neural Information Processing Systems, 37:74461– 74486, 2024.
>
> >The pipeline Figure 1 could be better organized and the Algorithm 1 could be expanded or separated for atomic presentaitons. Adding clearer subheadings and workflow guidance would improve readability and help convey the method’s structure more effectively.
>
> Thank you for the reviewer’s suggestion. We will refine Figure 1 to make the overall pipeline more aesthetically clear and intuitive.
>
> Regarding Algorithm 1, we have improved the layout while still keeping the two-column structure due to page limitations. Specifically, we added horizontal separators to clearly divide different stages of the algorithm, making the overall structure easier to follow. In addition, we introduced numbered subheadings for each stage, which helps clarify the flow of the method and reduces potential confusion.

---

> ### Author Response · Authors · 2025-11-21
> **Responses to reviewers: Question 1**
>
> >Since the method is built upon a Tensor-Train (TT) framework with efficient quantization, I suggest that the authors compare the inference cost with existing diffusion-based adversarial purification (DBP) methods, since DBP methods have been well explored as a strong purification paradigm. Such a comparison could better highlight the efficiency advantages of the proposed AP method.
>
> Thank you for the helpful suggestion.
>
> Following the reviewer’s recommendation, we evaluated the efficiency of our method by measuring both inference time and the amount of computation involved during inference. We compared our approach with a representative diffusion-based purification method (DiffPure) and a tensor-network–based purification baseline (TNP). All measurements were conducted on a single CIFAR-10 image using an NVIDIA RTX A6000 GPU.
>
> - **Inference time.**
>
> As shown in Table as follow, the results show that our approach is faster than DiffPure [1], and it also avoids the high computational cost of adversarial training. Compared with TNP [2], our method is slightly slower, mainly due to the additional optimization introduced by the wavelet-based regularization in our reconstruction process. This step provides the robustness benefits discussed in the paper, but also adds some overhead.
>
> | Methods  | DiffPure | TNP  | Ours |
> |----------|----------|------|------|
> | Time (s) | 6.86     | 1.25 | 1.75 |
>
> - **Inference parameters.**
>
> To further quantify efficiency, we also compared the number of parameters involved in the inference process for the three methods. As shown in Table as follow, diffusion-based purification requires orders of magnitude more parameters due to the heavy denoising backbone. In contrast, both tensor-network–based methods (TNP and ours) require significantly fewer parameters, with our method using the fewest among all three. This demonstrates that our approach maintains the lightweight nature of tensor-network–based purification while providing improved robustness.
>
> | Methods  | DiffPure | TNP  | Ours |
> |----------|----------|------|------|
> | Parameters |  143, 111, 773  | 34, 369 | 13, 756 |
>
> These results prove that our method achieves competitive robustness while remaining computationally efficient compared to diffusion-based purification and comparable tensor-network baselines.
>
> [1] Weili Nie et al.. Diffusion models for adversarial purification. arXiv preprint arXiv:2205.07460, 2022.
>
> [2] Guang Lin et al.. Model-free adversarial purification via coarse-to-fine tensor network representation. arXiv preprint arXiv:2502.17972, 2025.

---

> ### Author Response · Authors · 2025-11-21
> **Responses to reviewers: Question 3, 4**
>
> >The evaluation is limited to AutoAttack. To more rigorously validate the method, it should be tested against stronger adaptive attacks, such as PGD+EOT and BPDA+EOT, as well as purification-specific adaptive attacks like Diff-PGD [1] and DiffAttack [2]. A broader evaluation under various adaptive settings would strengthen the empirical credibility of the proposed defense.
>
> Thank you for your suggestion to include adaptive attacks. Following the recommendation, we additionally evaluated our method using Diff-PGD. The attack was configured according to the settings as follow:
>
> diffusion model accelerator: 50
>
> reverse steps in SDEdit: 2
>
> $l_\infty$ PGD budget: 8/255
>
> PGD iterations: 2
>
> The resulting performance is shown in Table as follow. The results show that, under settings where the clean accuracy is matched, our method achieves approximately 6\% higher robust accuracy compared to the TNP baseline.
> | Method | PSNR (Clean / Adv) | SSIM (Clean / Adv) | NRMSE (Clean / Adv) | Clean Acc. | Robust Acc. |
> |--------|----------------------|----------------------|-----------------------|-------------|--------------|
> | Ours   | 32.99 / **31.82**    | 0.8975 / **0.8570**  | 0.0486 / **0.0540**   | 64.00       | **40.00**    |
> | TNP    | 31.10 / **30.32**    | 0.9040 / **0.8609**  | 0.0574 / **0.0618**   | 65.20       | **34.00**    |
>
> Regarding DiffAttack, we attempted to reproduce the attack following the official implementation. However, DiffAttack requires a specific diffusion model checkpoint that the authors used, and this checkpoint is no longer available on HuggingFace. Since the attack is tightly coupled to that particular diffusion model, it is currently impossible to faithfully reproduce DiffAttack under the original protocol.
>
> The experiments for PGD+EOT and BPDA+EOT are currently in progress, and we will update the results in the supplementary material as soon as they are completed.
>
> >More implementation details are needed, including the tensor-network configurations, AutoAttack parameters, and the design of the adaptive attack setup used to evaluate the defense.
>
> Thank you for the reviewer’s suggestion.
>
> **Tensor-network configuration.**
>
> Our rank schedule follows the principle that the coarsest level contains the dominant low-frequency structure of the image, which is least affected by adversarial perturbations. Therefore, at this level, the rank is typically chosen to match the spatial resolution. As the pyramid progresses to finer scales, the residual becomes increasingly high-frequency and contains stronger adversarial noise; hence, we use ranks smaller than the corresponding resolution to constrain the expressive capacity of the reconstruction model.
>
> The exact resolutions and ranks used in our experiments are:
>
> ImageNet:
>
> Downsampled resolutions:
> [256, 128, 64, 32, 16]
>
> Corresponding QTT ranks:
> [29, 30, 31, 32, 16]
>
> CIFAR-10 / CIFAR-100:
>
> Downsampled resolutions:
> [64, 32, 16, 8]
>
> Corresponding QTT ranks:
> [4, 10, 16, 8]
>
> The regularization strength for the wavelet-based residual regularization is set to:
> $\lambda$ = 1e-6.
>
> **AutoAttack configuration.**
>
> Our AutoAttack settings strictly follow the RobustBench protocol, using its standard norm budgets for each dataset:
>
> CIFAR-10 / CIFAR-100 ($_\infty$): $\epsilon$ = 8/255
>
> CIFAR-10 ($_2$): $\epsilon$ = 0.5
>
> ImageNet ($_\infty$): $\epsilon$ = 4/255
>
> We use the white-box components of AutoAttack Standard without modification to hyperparameters.

---

> ### Author Response · Authors · 2025-11-21
> **Responses to reviewers: Question 5**
>
> >It would be helpful to include a visual comparison or supporting evidence demonstrating that coarse-to-fine tensor networks [3] tend to recover adversarial perturbations more frequently than the proposed method. Since the work is inspired from the previous findings[3].
>
> Thank you for the reviewer’s suggestion.
>
> Following the request, we conducted a visual comparison between the coarse-to-fine tensor-network reconstruction (PUTT) and our method. The results are shown in **Figure 8 in the revised version**. The left side shows the results from PUTT, and the right side shows our results. For each method, we present three images: the clean input (left), the reconstruction of the adversarial input (mid), and the pixel-wise difference between the two (right).
>
> We observe a clear qualitative distinction between the two approaches. In PUTT, adversarial perturbations remain visible across both smooth and textured regions, and the difference map exhibits **noise spread throughout the entire image**. This indicates that the coarse-to-fine TT reconstruction tends to retain adversarial noise at multiple scales, consistent with the high flexibility of TT cores in fine-resolution stages.
>
> In contrast, our method suppresses perturbations more effectively. The difference map shows that the remaining **variations are concentrated mainly around edges and high-frequency structures**, while large smooth regions are largely free of residual noise. This behavior aligns with our design: the wavelet-based regularization penalizes high-frequency sub-bands and prevents the reconstruction process from absorbing perturbations into the residual components.
>
> These visual results provide supporting evidence that, although inspired by PUTT, our method incorporates additional mechanisms that mitigate the tendency of TT-based reconstruction to recover adversarial perturbations.

---

> ### Author Response · Authors · 2025-11-26
> **Responses to reviewers: Question 2**
>
> >For each input resolution, the method requires training
>  separate tensor networks. While this design accounts for sensitivity to resolution, it substantially increases the training cost. Moreover, datasets such as CIFAR-10 and ImageNet differ greatly in resolution, leading to varied pooling depths and numbers of tensor networks, which may limit generalization to downstream tasks. Could the authors discuss whether potentially unifying the tensor-network architecture could significantly improve the overall soundness and scalability of the method?
>
> Thank you for the helpful comment!
>
> We would like to clarify that our method **does not train or maintain multiple tensor networks**. The pipeline carries out only **one QTT decomposition** at the beginning, which converts the input image into its low-rank tensor-train representation. Then, we update the data stored in the same QTT representation by reconstructing the image from the current tensor cores and minimizing the reconstruction loss relative to the target image at each stage. Thus, all optimization takes place within a single QTT, and the computational pathway does not involve multiple independent QTT decompositions.
>
> Regarding the dependence on input resolution, the number of QTT decompositions naturally follows from the multi-resolution structure of the purification pipeline. Datasets such as CIFAR-10 and ImageNet differ in resolution, so the number of pooling levels changes accordingly.
>
> **Designing a unified architecture is a topic worthy of further research.** If the tensor network were used purely as a decomposition tool, then a unified architecture could indeed be applied across different datasets. However, when the tensor network is used as a purification module, its role is not only to represent the image but also to suppress adversarial perturbations. Unlike natural image structures, adversarial perturbations are generated adaptively with respect to both the image content and the model’s gradients, and therefore their characteristics vary significantly across datasets of different styles, resolutions compositions.
>
> As a result, the structure and strength of the perturbations are not dataset-independent, and a single fixed configuration is unlikely to handle all cases effectively. In practice, the most reliable strategy is to adjust the tensor-network settings (such as rank scheduling and decomposition depth) to match the characteristics of each dataset. We agree that this is a promising direction, and we consider it an important avenue for future work.

---

> ### Author Response · Authors · 2025-11-27
> **Responses to reviewers: Question 3 (Supplement)**
>
> >The evaluation is limited to AutoAttack. To more rigorously validate the method, it should be tested against stronger adaptive attacks, such as PGD+EOT and BPDA+EOT, as well as purification-specific adaptive attacks like Diff-PGD [1] and DiffAttack [2]. A broader evaluation under various adaptive settings would strengthen the empirical credibility of the proposed defense.
>
> Regarding the reviewer’s request to evaluate the defense under PGD-EOT and BPDA-EOT, we would like to clarify that our purification pipeline contains several inherently non-differentiable components: multi-scale downsampling, tensor-network–based reconstruction (which itself is an iterative optimization procedure), and multi-scale upsampling. As a consequence, the entire process is non-differentiable, making full gradient attacks unable to obtain meaningful gradients to effectively attack our method.
>
> Therefore, following prior adaptive attack practices [1], we adopt BPDA as the adaptive attack strategy. During the backward pass, the purification module is approximated as an identity mapping. Consistent with the settings in [2], we use 200 PGD iterations and 20 EOT samples during the attack. We compared several AP/AT methods, the experimental results are reported below.
> | Type | Method        | Clean Acc. | Robust Acc. |
> |------|---------------|------------|-------------|
> | AT   | Gowal et al. [3]  | 87.51      | 66.01   |
> |      | Gowal et al. [4] | 88.71      | 65.93  |
> |      | Pang et al. [5]     | 88.62      | 64.95   |
> |------|---------------|------------|-------------|
> | AP   | Yoon et al. [6]  | 85.66      | 33.48   |
> |      | Nie et al. [7]   | 89.02      | 46.84   |
> |      | Lee et al. [2]   | 91.99      | 55.82   |
> |      | Ours          | 90.03      | 68.71   |
>
> [1] Guang Lin, Chao Li, Jianhai Zhang, Toshihisa Tanaka, and Qibin Zhao. Adversarial training on purification (atop): Advancing both robustness and generalization. In The Twelfth ICLR, 2024.
>
> [2] Minjong Lee and Dongwoo Kim. Robust evaluation of diffusion-based adversarial purification. In Proceedings of the ICCV, pp. 134–144, October 2023.
>
> [3] Sven Gowal, Chongli Qin, Jonathan Uesato, Timothy Mann, and Pushmeet Kohli. Uncovering the limits of adversarial training against norm-bounded adversarial examples. arXiv preprint:2010.03593, 2020.
>
> [4] Sven Gowal, Sylvestre-Alvise Rebuffi, Olivia Wiles, Florian Stimberg, Dan Andrei Calian, and Timothy A Mann. Improving robustness using generated data. Advances in Neural Information Processing Systems, 34:4218–4233, 2021.
>
> [5] Tianyu Pang, Min Lin, Xiao Yang, Jun Zhu, and Shuicheng Yan. Robustness and accuracy could be reconcilable by (proper) definition. In International Conference on Machine Learning, pp. 17258–17277. PMLR, 2022.
>
> [6] Jongmin Yoon, Sung Ju Hwang, and Juho Lee. Adversarial purification with score-based generative models. In International Conference on Machine Learning, pp. 12062–12072. PMLR, 2021.
>
> [7] Weili Nie, Brandon Guo, Yujia Huang, Chaowei Xiao, Arash Vahdat, and Anima Anandkumar. Diffusion models for adversarial purification. arXiv preprint arXiv:2205.07460, 2022.

---

> ### Author Response · Authors · 2025-11-30
> **Responses to reviewers: The revised version has been uploaded**
>
> Dear reviewers,
>
> We sincerely thank all reviewers for their thorough reading and highly insightful comments. We have carefully considered every suggestion and have revised the manuscript accordingly. A fully updated version has now been uploaded. For ease of verification, all modifications are in blue in the revised manuscript.
>
> We very much appreciate the reviewers’ professional feedback, and we look forward to any further comments and discussion.
>
> Best regards

---

### Official Review · Reviewer_baTF · 2025-10-29

**Soundness:** 3
**Presentation:** 4
**Contribution:** 3
**Rating:** 4
**Confidence:** 4

**Summary:**

To address the inherent conflict between reconstruction fidelity and denoising robustness in tensor-network-based defenses, the paper proposes a Progressive Residual Tensor Network for adversarial purification. The core idea is to introduce a Laplacian pyramid–inspired progressive residual reconstruction that decomposes the image into frequency bands and reconstructs residuals across scales. Specifically, at coarse levels, the model captures global semantics with higher rank capacity, while finer levels use decreasing ranks to suppress high-frequency adversarial residues. In addition, wavelet-based residual regularization (WRR) penalizes high-frequency energy, discouraging adversarial noise amplification. Extensive experiments on CIFAR-10, CIFAR-100, and ImageNet demonstrate superior robustness and competitive clean accuracy compared to state-of-the-art adversarial training and purification baselines.

**Strengths:**

1.	The design of progressive residual reconstruction and wavelet-based regularization is valuable, offering a frequency-aware solution to the reconstruction–denoising conflict in tensor-based defenses.
2.	The paper provides a clear motivation grounded in frequency-domain analysis and prior Laplacian pyramid theory, which justifies the proposed multi-scale residual modeling.
3.	The proposed method shows superior robustness and competitive clean accuracy across various datasets, exhibiting strong practicality.

**Weaknesses:**

1.	More design details about the rank schedule are needed. Although the experiments in Table 2 preliminarily compared the performance of different reconstruction settings, a more detailed design of the decreasing rank schedule (e.g., how to select and tune the value, interval, and corresponding resolution of ranks) remains heuristic. Providing analytical or empirical guidance for rank selection will enhance reproducibility.
2.	A broader range of attack strategies should be considered. Specifically, this study evaluates the proposed method only under AutoAttack. It would be beneficial to include analyses or experiments involving more recent and sophisticated attacks that demonstrate stronger capabilities in bypassing purification-based defenses, such as BPDA [A] and DiffHammer [B].
3.	A more detailed description of the white-box setting used in the proposed method is necessary. Specifically, AutoAttack comprises multiple modes, including both black-box and white-box attacks. It remains unclear how the proposed purification module is exposed to white-box adversaries. Since this aspect critically influences the validity of the robustness evaluation, a clearer and more thorough explanation is required.
4.	Lack of analysis on computational overhead. Specifically, the paper does not sufficiently discuss the training and inference cost of multi-stage tensor reconstruction, which may be substantial given the multiple QTT decompositions per resolution level. A runtime or memory comparison against prior works would be valuable.
5.	In addition to adversarial purification and adversarial training, adversarial detection methods [C] [D] [E] also achieve defense against attack samples. It would be valuable for the authors to further discuss and compare these approaches, highlighting their similarities and distinctions relative to the defense strategy proposed in this work.

**Minor issues**

1.	The pseudo-code can be reformatted into a consistent two-column layout. The current presentation is hard to read.

2.	There is a figure index error in section 4.4.

**Questions:**

None

---

> ### Author Response · Authors · 2025-11-21
> **Responses to reviewers: Weakness 1, 2**
>
> >More design details about the rank schedule are needed. Although the experiments in Table 2 preliminarily compared the performance of different reconstruction settings, a more detailed design of the decreasing rank schedule (e.g., how to select and tune the value, interval, and corresponding resolution of ranks) remains heuristic. Providing analytical or empirical guidance for rank selection will enhance reproducibility.
>
> We sincerely thank the reviewer for the careful evaluation of our work and the insightful suggestions!
>
> Our rank selection is not arbitrary but follows a clear structural logic dictated by pyramid decomposition.
>
> **At the coarsest level**, the reconstruction target encompasses the image's low-frequency components, capturing dominant semantic structures. Downsampling severely degrades adversarial perturbations, rendering this component “clean.” Consequently, this level is assigned a large rank value (typically equal to the coarse feature map's resolution) to enable the Tensor Network to fully reconstruct critical global information.
>
> **As resolution increases**, residual components gradually become sparse and high-frequency, with corresponding intensification of adversarial perturbations. To prevent overfitting in these fine-grained stages and preserve perturbations, the expressive power of the tensor network is deliberately constrained by adopting smaller rank values (typically far below the corresponding resolution). This rank selection strategy controls the model's flexibility in regions where adversarial noise is most pronounced.
>
> Regarding rank-selection methodology, we agree that adaptive rank adjustment is a promising future direction. However, the main challenge lies in the difficulty of reliably modeling adversarial perturbations: their spectral behavior is highly image-dependent. Designing a mechanism that simultaneously balances reconstruction fidelity, perturbation suppression, and rank adjustment remains an open research problem. **In this work, we therefore adopt a fixed resolution-dependent schedule, where ranks are chosen based on the best empirical performance per dataset.** Developing principled automatic rank selection is one of our planned future extensions.
>
> >A broader range of attack strategies should be considered. Specifically, this study evaluates the proposed method only under AutoAttack. It would be beneficial to include analyses or experiments involving more recent and sophisticated attacks that demonstrate stronger capabilities in bypassing purification-based defenses, such as BPDA [A] and DiffHammer [B].
>
> We thank the reviewer for this valuable suggestion.
>
> We are currently evaluating our method under BPDA-based adaptive attacks, and the experiments are in progress. We will include the complete results in the updated supplementary material.
>
> Regarding DiffHammer, we note that this attack is designed explicitly for diffusion-based purification pipelines. Since our method does not employ any diffusion model or sampling-based purification, the core assumptions underlying DiffHammer do not hold, and the attack is not applicable to our method.

---

> ### Author Response · Authors · 2025-11-21
> **Responses to reviewers: Weakness 3**
>
> >A more detailed description of the white-box setting used in the proposed method is necessary. Specifically, AutoAttack comprises multiple modes, including both black-box and white-box attacks. It remains unclear how the proposed purification module is exposed to white-box adversaries. Since this aspect critically influences the validity of the robustness evaluation, a clearer and more thorough explanation is required.
>
> Thank you for pointing out the need for a clearer description of the white-box setting. In a white-box scenario, the adversary is assumed to have full access to the entire purification pipeline, including the tensor decomposition and reconstruction steps. However, **tensor-network–based purification module does not contain any trainable parameters**. The tensor network operates purely as an optimization procedure: it decomposes each input image into a low-rank representation that is specific to that input, rather than relying on learned weights. As a result, unlike standard neural-network–based defenses, the module does not expose stable gradients or trainable parameters that an attacker could exploit. Therefore, gradient-based white-box attacks cannot effectively target the tensor network itself.
>
> In addition, we are currently running PGD+EOT and BPDA+EOT experiments to further expose the purification module to strong adaptive adversaries. The complete results and the detailed attack setup will be included in an updated supplementary version.

---

> ### Author Response · Authors · 2025-11-21
> **Responses to reviewers: Weakness 4**
>
> >Lack of analysis on computational overhead. Specifically, the paper does not sufficiently discuss the training and inference cost of multi-stage tensor reconstruction, which may be substantial given the multiple QTT decompositions per resolution level. A runtime or memory comparison against prior works would be valuable.
>
> We appreciate the reviewer’s suggestion. Although the main focus of the paper is the robustness behavior of the proposed reconstruction mechanism, we agree that reporting computational overhead is important for completeness.
>
> We would like to clarify that our method **does not repeatedly perform QTT decompositions** at every resolution level. The pipeline carries out only one QTT decomposition at the beginning, which converts the input image into its low-rank tensor-train representation. Then, we update the data stored in the same QTT representation by reconstructing the image from the current tensor cores and minimizing the reconstruction loss relative to the target image at each stage. Thus, all optimization takes place within a single QTT, and the computational pathway **does not involve multiple independent QTT decompositions**.
>
> Unlike neural network-based defense mechanisms or adversarial training, tensor network-based methods involve no learnable parameters. QTT decomposition extracts core features from input data through low-rank decomposition. Consequently, our approach requires no training phase, thereby avoiding the substantial computational overhead associated with adversarial training.
>
> We have added a simple **complexity analysis** and also measured the actual **inference time** and the amount of **inference parameters** during the inference process on Cifar10.
>
> - **Complexity analysis.**
>   We added a new subsection that analyzes in detail the computational complexity of each component in our implementation pipeline, including:
>     (1) multi-scale image operations (upsampling and pooling),
>     (2) QTT decomposition and reconstruction,
>     (3) MPO-based prolongation,
>     (4) TT-SVD rank truncation,
>     (5) residual update, and
>     (6) final multi-scale synthesis.
>
>     Let the spatial resolution at level $k$ be $H_k = 2^k$ with $C$ channels, and denote the maximum TT-rank by $r_k \le r_{\max}$.
>     The computational cost at level $k$ becomes
>
>     $\mathcal{C}_k=I_k\times[\mathcal{O}(2^{2k} C)+\mathcal{O}(k r_k^2 2^{2k})]+\mathcal{O}(k r_k^3),$
>
>     where $I_s$ is the number of training iterations and the three terms correspond respectively to 2D image operations, QTT reconstruction, and MPO + TT-SVD rank truncation.
>
>     Since $H_k$ increases exponentially with $k$, the total complexity is ultimately dominated by the highest resolution, $H_{D} = 2^D$. The total cost can therefore be approximated as
>
>    $\mathcal{C_D} =I_D\times[\mathcal{O}(H^{2}_{D} C)+\mathcal{O}(r^2_D H^2_D log(H_D))]$
>
>    The first term reflects the cost of multi-scale 2D image processing, while the second term arises from QTT-based reconstruction.
>
> - **Inference time.**
>
>     We also tested a representative Diffusion-based AP method (DiffPure) [1] and a tensor-based method (TNP) [2]. The results are presented in Table as follow. The result shows that our method is significantly faster than DiffPure, which require dozens of sampling steps, and is comparable to TNP. Our current implementation still has room for improvement, and making the reconstruction module faster is one of our priorities for future work.
>
>    | Methods  | DiffPure | TNP  | Ours |
>    |----------|----------|------|------|
>    | Time (s) | 6.86     | 1.25 | 1.75 |
>
> - **Inference parameters.**
>    To further quantify efficiency, we also compared the number of parameters involved in the inference process for the three methods. As shown in Table as follow, DiffPure requires orders of magnitude more parameters due to the heavy denoising backbone. In contrast, both tensor-network–based methods (TNP and ours) require significantly fewer parameters, with our method using the fewest among all three. This demonstrates that our approach maintains the lightweight nature of tensor-network–based purification while providing improved robustness.
>
>    | Methods  | DiffPure | TNP  | Ours |
>    |----------|----------|------|------|
>    | Parameters |  143, 111, 773  | 34, 369 | 13, 756 |
>
> [1] Weili Nie, Brandon Guo, Yujia Huang, Chaowei Xiao, Arash Vahdat, and Anima Anandkumar. Diffusion models for adversarial purification. arXiv preprint arXiv:2205.07460, 2022.
>
> [2] Guang Lin, Duc Thien Nguyen, Zerui Tao, Konstantinos Slavakis, Toshihisa Tanaka, and Qibin Zhao. Model-free adversarial purification via coarse-to-fine tensor network representation. arXiv preprint arXiv:2502.17972, 2025.

---

> ### Author Response · Authors · 2025-11-21
> **Responses to reviewers: Weakness 5 and Minor issues**
>
> >In addition to adversarial purification and adversarial training, adversarial detection methods [C] [D] [E] also achieve defense against attack samples. It would be valuable for the authors to further discuss and compare these approaches, highlighting their similarities and distinctions relative to the defense strategy proposed in this work.
>
> We appreciate the reviewer’s suggestion. Adversarial detection (AD) methods indeed form an important branch of adversarial robustness research, but their objective and usage scenario differ fundamentally from adversarial purification (AP), and thus their goals differ sufficiently that a one-to-one comparison does not naturally apply.
>
> AD methods aim to decide whether an input is adversarial or clean. When an input is flagged as adversarial, the system typically rejects the sample and does not perform classification. This restricts their applicability to settings where rejection is acceptable and where downstream systems can safely ignore or defer prediction.
>
> In contrast, our work follows the AP paradigm, whose goal is entirely different: to reconstruct a clean version of the input so that every sample—clean or adversarial—can still be processed by the classifier. AP methods do not reject inputs; they attempt to restore them. As a result, AP and AD operate under different constraints and solve different problems:
>
> For AP, success means recovering a usable, clean input.
>
> For AD, success means detecting and rejecting harmful inputs.
>
> Because AD methods depend on thresholding mechanisms, acceptance criteria, and system-level policies, while AP focuses on signal reconstruction quality and robustness, their design philosophies are not directly comparable.
>
>
> >Minor issues:
> 1.The pseudo-code can be reformatted into a consistent two-column layout. The current presentation is hard to read.
> 2.There is a figure index error in section 4.4.
>
> Thank you very much for the careful reading of our manuscript and for pointing out these minor presentation issues. We will correct the figure index error in Section 4.4.
>
> Regarding the pseudo-code, we agree that the current layout can be improved for better readability. Due to space limitations in the main paper, we will revise the pseudo-code formatting and move the full version to the supplementary material to ensure a cleaner and more readable presentation.

---

> ### Author Response · Authors · 2025-11-30
> **Responses to reviewers: The revised version has been uploaded**
>
> Dear reviewers,
>
> We sincerely thank all reviewers for their thorough reading and highly insightful comments. We have carefully considered every suggestion and have revised the manuscript accordingly. A fully updated version has now been uploaded. For ease of verification, all modifications are in blue in the revised manuscript.
>
> We very much appreciate the reviewers’ professional feedback, and we look forward to any further comments and discussion.
>
> Best regards

---

### Official Review · Reviewer_LvRB · 2025-10-31

**Soundness:** 3
**Presentation:** 2
**Contribution:** 3
**Rating:** 6
**Confidence:** 3

**Summary:**

Authors propose a new adversarial purification method based on Feature Pyramid Networks (FPN) that progressively recovers the clean component. Each level filters a distinct frequency band, so layers do not interfere with each other. A Haar wavelet–based regularizer is applied at every layer to trim the high‑frequency domain at each resolution level.

**Strengths:**

1. The architecture of the progressive residual reconstruction is clearly explained and overall reasonably motivated.
2. The usage of Wavelet-Based Residual Regularization is supported by Figure 2a.

**Weaknesses:**

1. The absence of reported time measurements is a significant limitation. Quantifying runtime efficiency is crucial for assessing the method's suitability for real-time applications and for clearly defining its practical use cases. I would also like to see a discussion about the method complexity.
2. The manuscript lacks a defined strategy for selecting the decomposition rank, as well as any method for rank adaptation. This is a noticeable oversight, as the optimal rank is likely dependent on image-specific factors such as resolution and detail complexity.
3. The evaluation is limited to a 2-level pyramid, and the rationale for this constraint is not provided. Clarification is needed on whether this choice was due to empirical performance plateaus, computational trade-offs, or degradation in output quality with additional levels.

**Questions:**

1. Could you clarify what is the reason behind using "sub-bands" over which you compute $\ell_1$ norm (P.6, eq.9)?
2. It would be beneficial to report compression ratios to better understand the method.
3. What is implied by "Strategy A" and "Strategy B" in Section 5.3?
4. Did you try wavelet transform separately from Tensor network purification?

Minor presentation issues:
- At the top of Figure 1 rank should be $r_{d-l}$ instead of $d-l$.
- $A^{1}$ is repeated in Figure 1.
- In Figure 1, the label near the right arrow should be $y_d$, not $x_d$.
- Placing algorithm in a two-column mode makes it more difficult to analyse.
- In the second column of the Algrotihm 1, $\mathcal{L}$ should have a subscript index.
- Table 2 does not have any dataset reference.

---

> ### Author Response · Authors · 2025-11-21
> **Responses to reviewers: Weakness 1**
>
> >The absence of reported time measurements is a significant limitation. Quantifying runtime efficiency is crucial for assessing the method's suitability for real-time applications and for clearly defining its practical use cases. We would also like to see a discussion about the method complexity.
>
> **Response to Reviewer.**
> Thank you for carefully reviewing our paper and for providing valuable suggestions. Below we provide point-by-point responses to the comments.
>
> We evaluated the runtime of our method and also conducted a detailed computational complexity analysis. The newly added content is summarized below.
>
> - **Inference time.**
>
>  We have measured the actual inference time on Cifar10. We also tested a representative Diffusion-based AP method (DiffPure) [1] and a tensor-based method (TNP) [2]. The results are presented in Table as follow. The result shows that our method is significantly faster than DiffPure, which require dozens of sampling steps, and is comparable to TNP. Our current implementation still has room for improvement, and making the reconstruction module faster is one of our priorities for future work.
>
> | Methods  | DiffPure | TNP  | Ours |
> |----------|----------|------|------|
> | Time (s) | 6.86     | 1.25 | 1.75 |
>
> - **Complexity analysis.**
>   We added a new subsection that analyzes in detail the computational complexity of each component in our implementation pipeline, including:
>     (1) multi-scale image operations (upsampling and pooling),
>     (2) QTT decomposition and reconstruction,
>     (3) MPO-based prolongation,
>     (4) TT-SVD rank truncation,
>     (5) residual update, and
>     (6) final multi-scale synthesis.
>
>     Let the spatial resolution at level $k$ be $H_k = 2^k$ with $C$ channels, and denote the maximum TT-rank by $r_k \le r_{\max}$.
>     The computational cost at level $k$ becomes
>
>     $\mathcal{C}_k=I_k\times[\mathcal{O}(2^{2k} C)+\mathcal{O}(k r_k^2 2^{2k})]+\mathcal{O}(k r_k^3),$
>
>     where $I_s$ is the number of training iterations and the three terms correspond respectively to 2D image operations, QTT reconstruction, and MPO + TT-SVD rank truncation.
>
>     Since $H_k$ increases exponentially with $k$, the total complexity is ultimately dominated by the highest resolution, $H_{D} = 2^D$. The total cost can therefore be approximated as
>
>    $\mathcal{C_D} =I_D\times[\mathcal{O}(H^{2}_{D} C)+\mathcal{O}(r^2_D H^2_D log(H_D))]$
>
>    The first term reflects the cost of multi-scale 2D image processing, while the second term arises from QTT-based reconstruction.
>
> [1] Weili Nie, Brandon Guo, Yujia Huang, Chaowei Xiao, Arash Vahdat, and Anima Anandkumar. Diffusion models for adversarial purification. arXiv preprint arXiv:2205.07460, 2022.
>
> [2] Guang Lin, Duc Thien Nguyen, Zerui Tao, Konstantinos Slavakis, Toshihisa Tanaka, and Qibin Zhao. Model-free adversarial purification via coarse-to-fine tensor network representation. arXiv preprint arXiv:2502.17972, 2025.

---

> ### Author Response · Authors · 2025-11-21
> **Responses to reviewers: Weakness 2**
>
> >The manuscript lacks a defined strategy for selecting the decomposition rank, as well as any method for rank adaptation. This is a noticeable oversight, as the optimal rank is likely dependent on image-specific factors such as resolution and detail complexity.}
>
> We appreciate the reviewers' insightful comments!
>
> The decomposition rank is indeed a crucial design parameter in tensor train models. We wish to clarify that the rank values employed in our method are not arbitrarily chosen but determined based on the characteristics of the reconstruction objective: higher ranks are required in the low-resolution/low-frequency stage to capture global structures, while lower ranks suffice for reconstruction in the high-resolution/high-frequency stage. This strategy achieves good reconstruction quality without recovering adversarial perturbations. Following this principle, we typically set the rank of the lowest-resolution stage equal to its corresponding resolution, while the ranks for higher-resolution stages are chosen to be smaller than their respective resolutions.
>
> Regarding adaptive rank selection, we agree that automatic rank adjustment is a promising future direction. However, due to the difficulty in effectively modeling adversarial perturbations, it is currently challenging to establish a rank estimation mechanism that balances image reconstruction, perturbation suppression, and rank selection. At present, we determine the rank based on the method's optimal experimental results achieved on the datasets. Developing an effective rank selection strategy remains one of our key focuses for future work.

---

> ### Author Response · Authors · 2025-11-21
> **Responses to reviewers: Weakness 3**
>
> >The evaluation is limited to a 2-level pyramid, and the rationale for this constraint is not provided. Clarification is needed on whether this choice was due to empirical performance plateaus, computational trade-offs, or degradation in output quality with additional levels.
>
> We apologize for the misunderstanding caused by the insufficient explanation in our manuscript!
>
> The two-level architecture was not the final design used in our method; it was only the configuration specific to the experiments in Table 2 of our paper, rather than the architecture adopted in the full model.
>
> Our use of a 2-level pyramid was not the result of tuning or selective reporting. Instead, **it reflects the minimal configuration needed to realize the core idea of our framework**, namely separating an image into distinct frequency bands and reconstructing them with different capacities. A single-level decomposition does not create meaningful subbands, whereas a 2-level Laplacian pyramid is the smallest structure that provides both a coarse (low-frequency) component and a residual high-frequency component, enabling the multi-band reconstruction mechanism that Laplacian pyramid (LP) + Wavelet-Based Residual Regularization (WRR) is designed to study.
>
> The purpose of Table 2 in our paper is to analyze how different reconstruction strategies trade off clean fidelity versus robustness, rather than to optimize the number of pyramid levels. For this analysis, the 2-level configuration already fully exposes the behavior we aim to investigate: the interactions between low-frequency reconstruction, high-frequency residual suppression, and the WRR regularizer.
>
> We added experiments with 3-layer and 5-layer pyramid structures. As the result s show int Table, deeper pyramid structures exhibit trends consistent with Level 2 structures: the trade-off between overall reconstruction accuracy and robustness remains largely unchanged. As the number of layers increases, our method achieves a better trade-off by suppressing noisy recoveries through rank constraints.
>
> ## **Direct Reconstruction**
> | Resolution | QTT Rank | PSNR (Clean / Adv) | SSIM (Clean / Adv) | NRMSE (Clean / Adv)     | Acc (%) (Clean / Adv) |
> | ---------- | -------- | ------------- | --------------- | --------------- | ------------- |
> | 256        | 50       | 31.97 / 31.72 | 0.8722 / 0.8610 | 0.0547 / 0.0560 | 65.79 / 48.00 |
>
> ## **Progressive Reconstruction**
> | Resolution | QTT Rank | PSNR (Clean / Adv) | SSIM (Clean / Adv) | NRMSE (Clean / Adv)     | Acc (%) (Clean / Adv) |
> | ---------- | -------------- | ------------- | --------------- | --------------- | ------------- |
> | 128→256    | 50-50          | 35.61 / 34.99 | 0.9211 / 0.9101 | 0.0387 / 0.0406 | 71.48 / 40.23 |
> | 64→256     | 50-50-50       | 35.58 / 34.97 | 0.9211 / 0.9100 | 0.0388 / 0.0406 | 71.60 / 39.90 |
> | 16→256     | 50-50-50-50-50 | 36.10 / 35.32 | 0.9240 / 0.9147 | 0.0367 / 0.0400 | 71.67 / 38.99 |
>
> ## **Laplacian Pyramid–Based Reconstruction**
> | Resolution | QTT Rank | PSNR (Clean / Adv) | SSIM (Clean / Adv) | NRMSE (Clean / Adv)     | Acc (%) (Clean / Adv) |
> | ---------- | -------------- | ------------- | --------------- | --------------- | ------------- |
> | 128→256    | 50-50          | 36.87 / 36.20 | 0.9410 / 0.9313 | 0.0326 / 0.0344 | 72.85 / 36.52 |
> | 64→256     | 50-30          | 33.36 / 32.98 | 0.8955 / 0.8848 | 0.0477 / 0.0492 | 67.96 / 48.39 |
> | 64→256     | 50-50-50       | 37.99 / 37.34 | 0.9558 / 0.9473 | 0.0281 / 0.0298 | 73.00 / 31.64 |
> | 16→256     | 50-50-50-50-50 | 38.44 / 37.58 | 0.9572 / 0.9484 | 0.0277 / 0.0295 | 74.44 / 29.66 |
> | 16→256     | 16-32-31-30-29 | 33.57 / 33.19 | 0.9032 / 0.8919 | 0.0472 / 0.0487 | 68.16 / 49.21 |

---

> ### Author Response · Authors · 2025-11-21
> **Responses to reviewers: Question 1-4**
>
> >Could you clarify what is the reason behind using "sub-bands" over which you compute $\ell_1$ norm (P.6, eq.9)?
>
> We appreciate the reviewers' insightful comments! We would like to note that this point is discussed in the paragraph following Equation (9) in the paper.
>
> Our motivation for applying the $l_1$ penalty over the sub-bands rather than in the pixel domain provides two additional benefits:
>
> **Selective control of frequency-direction components.**
> The LH, HL, and HH sub-bands correspond to horizontal, vertical, and diagonal high-frequency structures. Computing the $l_1$ norm over these bands allows us to selectively constrain the portions of the representation where perturbations most frequently accumulate. Applying an $l_1$ penalty directly to the image space would suppress both noise and meaningful fine-scale structure indiscriminately, whereas sub-band decomposition cleanly separates where regularization should be applied.
>
> **Stabilizing progressive reconstruction.**
> Because the high-resolution stages are responsible for adding fine structural details, regularizing their sub-band coefficients prevents them from over-amplifying spurious high-frequency signals. This aligns with our intention stated in the paper: WRR stabilizes reconstruction by ensuring that high-resolution stages contribute only necessary semantic detail rather than absorbing adversarial noise.
>
> >It would be beneficial to report compression ratios to better understand the method.
>
> For the **ImageNet** dataset, we upsample each image to a spatial resolution of 256×256, and the compression ratios of the 5-level QTT decomposition at coarse-to-fine resolutions [16, 32, 64, 128, 256] are as follows: [0.65, 0.77, 1.59, 4.48, 14.29].
>
> For the **Cifar10** dataset, we upsample each image to 64×64 and the compression ratios of the 4-level QTT decomposition at coarse-to-fine resolutions [8, 16, 32, 64] are as follows: [0.8, 0.69, 2.8, 38.4].
>
> These values reflect how the QTT representation becomes increasingly compact at finer scales, illustrating the efficiency of the QTT structure across different resolutions.
>
> >What is implied by "Strategy A" and "Strategy B" in Section 5.3?
>
> Thank you for pointing this out.
>
> In Section 5.3, “Strategy A”, “Strategy B”, and “Strategy C” refer respectively to (A) direct reconstruction, (B) progressive reconstruction, and (C) Laplacian-pyramid–based reconstruction. These labels were introduced merely to simplify notation and improve the readability of Table 2.
>
> We agree that the manuscript should make this correspondence more explicit to avoid confusion. In the revised version, we will remove these three terms to make the paper cleaner and easier to read.
>
> >{Did you try wavelet transform separately from Tensor network purification?
>
> We appreciate the reviewers' insightful comments!
>
> We additionally evaluated a baseline approach using only wavelet-based purification. To ensure consistency in experimental setup, input images were decomposed via a 4-level Haar discrete wavelet transform (DWT). High-frequency subbands (LH/HL/HH) underwent soft thresholding before image reconstruction through inverse discrete wavelet transform (i.e., without employing tensor networks).The results on ImageNet are as follows:
>
> | Method      | PSNR (Clean / Adv)   | SSIM (Clean / Adv)     | NRMSE (Clean / Adv)    | Clean Acc. | Robust Acc. |
> |-------------|------------------------|---------------------------|--------------------------|--------------|---------------|
> | 4-level DWT | 33.71 / **32.42**      | 0.8945 / **0.8945**       | 0.0404 / **0.0467**      | 68.55        | **39.65**     |
> | Ours        | 33.57 / **33.19**      | 0.9032 / **0.8919**       | 0.0472 / **0.0487**      | 68.16        | **49.21**     |
>
> Compared to pure wavelet thresholding methods, our approach achieves a 10\% improvement in robust accuracy while maintaining equivalent clean accuracy.

---

> ### Author Response · Authors · 2025-11-21
> **Responses to reviewers: Minor presentation issues**
>
> >Minor presentation issues:
> - At the top of Figure 1 rank should be $r_{d-l}$ instead of $d-l$.
> - $A^{1}$ is repeated in Figure 1.
> - In Figure 1, the label near the right arrow should be $y_d$, not $x_d$.
> - Placing algorithm in a two-column mode makes it more difficult to analyse.
> - In the second column of the Algrotihm 1, $\mathcal{L}$ should have a subscript index.
> - Table 2 does not have any dataset reference.
>
> We thank the reviewer for carefully identifying these presentation issues! We will correct all of them in the revised manuscript.

---

> ### Author Response · Authors · 2025-11-30
> **Responses to reviewers: The revised version has been uploaded**
>
> Dear reviewers,
>
> We sincerely thank all reviewers for their thorough reading and highly insightful comments. We have carefully considered every suggestion and have revised the manuscript accordingly. A fully updated version has now been uploaded. For ease of verification, all modifications are in blue in the revised manuscript.
>
> We very much appreciate the reviewers’ professional feedback, and we look forward to any further comments and discussion.
>
> Best regards

---

### Official Review · Reviewer_UAmz · 2025-10-31

**Soundness:** 2
**Presentation:** 2
**Contribution:** 2
**Rating:** 2
**Confidence:** 4

**Summary:**

Proposes a Laplacian pyramid–inspired progressive residual reconstruction for tensor-network adversarial purification, with resolution-dependent rank scheduling and a Haar wavelet–based residual regularization (WRR) on high-frequency subbands. Evaluated on CIFAR-10/100 and ImageNet under AutoAttack, with comparisons to AT/AP baselines.

**Strengths:**

- Clear articulation of the reconstruction-denoising conflict and a structured attempt to address it via scale separation and capacity control.
- Progressive residual formulation is simple, potentially broadly applicable, and compatible with existing classifiers.
- Includes comparisons to several AT/AP methods and a basic reconstruction-strategy ablation.

**Weaknesses:**

- Justification of Laplacian choice is insufficient: the paper motivates LP historically, but does not compare to alternative pyramids such as Gaussian pyramids or wavelet pyramids, nor to multi-level wavelet reconstructions that more directly align with the WRR penalty. The only reconstruction-strategy ablation is Table 2 (direct vs progressive vs LP) with limited scope.

- Ablations are thin: Table 2 is the sole substantive ablation; there is no study varying the decomposition family, number of scales, or alternative rank policies beyond a few QTT rank pairs.

- Subband handling is oversimplified: WRR penalizes the concatenated high-frequency bands, but no per-band analysis (LH/HL/HH individually or pairwise) is provided, despite claims about “high-frequency” components; Fig. 2a labels “high vs low”, while Haar yields LL, LH, HL, HH.

- Trade-off instability: In Table 2, settings that improve robust accuracy tend to hurt clean (IID) accuracy and vice-versa, and the method without a robust backbone trails strong AT in some settings; the paper does not resolve when LP+WRR is preferable.

- Protocol clarity: Table 1 aggregates results across datasets with different classifier backbones (ResNet-50 vs WideResNet-28-10), complicating method-to-method comparison. QTT rank settings used for headline Table-1 numbers are not explicit.

- Baseline coverage and consistency: Although several AT/AP baselines are listed in Table 1, alignment with common RobustBench protocols is unclear, and baseline choices/backbones differ across datasets, reducing comparability.

- Missing related work: No mention of FLC/anti-aliasing pooling literature (e.g., [1]) that is relevant to frequency-aware robustness and could inform design choices.

References:
[1] Grabinski, Julia, et al. "Frequencylowcut pooling-plug and play against catastrophic overfitting." European Conference on Computer Vision. Cham: Springer Nature Switzerland, 2022.

**Questions:**

- Decomposition choice: Why Laplacian pyramid over Gaussian or wavelet pyramids? Could multi-level wavelet reconstruction subsume LP+WRR?

- Subband granularity: WRR aggregates LH/HL/HH. What happens if only HH is penalized, or LH/HL individually, or LH+HL without HH? Any per-band sensitivity that supports the “high-frequency energy” claim beyond the aggregate.

- Headline settings: Which QTT rank/scale settings and WRR λ were used for the headline Table-1 results on each dataset?

- Backbone consistency: Why switch to WideResNet-28-10 on CIFAR-100 while using ResNet-50 elsewhere. Can the main table be normalized to a single backbone per dataset for fair comparison?

- Benchmark alignment: Were the AT baselines reproduced under RobustBench-style evaluation, or are they literature numbers with heterogeneous training data/regularization. If the latter, can a standardized subset be added?

---

> ### Author Response · Authors · 2025-11-21
> **Responses to reviewers: Weakness 1 and Question 1**
>
> Thank you for carefully reviewing our paper and for providing valuable suggestions. Below we provide point-by-point responses to the comments.
>
> We would like to clarify that we did not choose the Laplacian pyramid (LP) because we considered it an optimal design. Rather, our intention was to construct a progressive reconstruction mechanism that learns image components across different frequency bands in a coarse-to-fine manner. This design naturally leads to a formulation mathematically equivalent to an LP, and thus we adopted the terminology as a clearer description. LP was not the motivation but a post-hoc equivalence.
>
> **Why not Gaussian Pyramids (GP)?**
> Gaussian pyramids repeatedly smooth the image, causing unavoidable loss of image details. This approach cannot satisfy the requirement of producing a clean and clear image.
>
> **Why not Wavelet Pyramids (WP)?**
> Wavelet pyramids use a fixed linear transform that predetermines the frequency split (LL/LH/HL/HH) and only further decomposes the LL band. Thus, the frequency partition is fixed from the start. In contrast, our method requires an adaptive residual decomposition whose structure is determined by optimization and regulated by rank scheduling.
>
> Thus, while multi-level wavelet reconstruction is a valid multi-resolution analysis tool, it addresses a different problem from ours: it provides a fixed transform basis, whereas our framework focuses on learning residuals that reflect the unexplained frequency bands under constrained tensor capacity. The Wavelet-Based Residual Regularization (WRR) penalty operates in the wavelet domain only as a regularizer for learned high-frequency components, not as a decomposition mechanism.
>
> To further clarify the distinction between the two approaches, we designed a controlled comparison experiment for ImageNet: we constructed a four-level wavelet transform to generate the reconstruction targets, with four levels chosen to match the number of reconstruction targets used in our method. As shown in the table, the wavelet-based variant performs worse than our method in both reconstruction quality and robustness metrics. This is because wavelet transforms merely filter high-frequency components with fixed bases, whereas our approach performs adaptive reconstruction and denoising tailored to the image content.
>
> We hope this resolves the misunderstanding: LP terminology is used solely as an accurate descriptor of the residual structure induced by our design, rather than as a normative architectural choice.
>
> | Method             | PSNR (Clean / Adv) | SSIM (Clean / Adv) | NRMSE (Clean / Adv) | Clean Acc. | Robust Acc. |
> |--------------------|--------------------|---------------------|----------------------|------------|--------------|
> | Laplacian Pyramid  | 33.57 / 33.19      | 0.9032 / 0.8919     | 0.0472 / 0.0487      | 68.16     | 49.21       |
> | Wavelet Pyramid   | 31.07 / 30.95      | 0.9196 / 0.9154     | 0.0471 / 0.0478      | 67.96     | 26.56       |

---

> ### Author Response · Authors · 2025-11-21
> **Responses to reviewers: Weakness 2**
>
> >Weakness 2: Ablations are thin: Table 2 is the sole substantive ablation; there is no study varying the decomposition family, number of scales, or alternative rank policies beyond a few QTT rank pairs.
>
> We thank the reviewer for the insightful comments!
>
> **Decomposition family:** Our choice of QTT is directly inspired by [1]. This work demonstrates that QTT models can progressively reconstruct an image by transitioning from a low-resolution representation to a higher-resolution one. This **coarse-to-fine** reconstruction mechanism is essential for obtaining high-quality tensor representations and naturally aligns with our goal of progressively recovering a clean image across pyramid levels.
>
> In contrast, classical tensor decompositions such as CP and Tucker do not possess such a structural mechanism for resolution expansion. Neither decomposition supports a principled or stable way to propagate a low-resolution representation to a higher-resolution one.
>
> For these structural reasons, QTT is suitable for our coarse-to-fine reconstruction framework, whereas CP and Tucker are fundamentally incompatible with the multiscale upsampling required by our method.
>
> **Number of scales:** We supplemented our experiments with more comprehensive ablation studies on scale, with results presented in Table. It can be observed that, at a fixed rank, as the scale increases, the image reconstruction performance gradually improves, leading to an increase in clean accuracy. However, the recovery of adversarial noise also becomes more pronounced, resulting in a decrease in robust accuracy. As the resolution decreases below 16, the model's reconstruction performance and purification effectiveness remain stable. We believe that excessively low resolutions cause the image to lose semantic information, thereby failing to further enhance the model's reconstruction capabilities.
>
> **Rank policies:** Adversarial perturbations are difficult to model and do not exhibit stable statistical or spectral patterns. Therefore, designing a principled rank-selection strategy that balances reconstruction quality and noise suppression remains a challenging open problem. At this stage, we determine rank values empirically based on the best experimental performance.
>
> [1] Sebastian Loeschcke, Dan Wang, Christian Leth-Espensen, Serge Belongie, Michael J Kastoryano, and Sagie Benaim. Coarse-tofine tensor trains for compact visual representations. In The Fortyfirst International Conference on Machine Learning, 2024.
>
> | Resolution | Rank              | PSNR (Clean / Adv) | SSIM (Clean / Adv) | NRMSE (Clean / Adv) | Clean Acc. | Robust Acc. |
> |-----------|-------------------|---------------------|---------------------|----------------------|------------|--------------|
> | 128→256   | 50,50             | 36.87 / 36.20       | 0.9419 / 0.9313     | 0.0326 / 0.0344      | 72.85      | 36.52        |
> | 64→256    | 50,50,50          | 37.99 / 37.34       | 0.9558 / 0.9473     | 0.0281 / 0.0298      | 73.00      | 31.64        |
> | 32→256    | 50,50,50,50       | 38.10 / 37.40       | 0.9560 / 0.9477     | 0.0280 / 0.0298      | 73.85      | 30.83        |
> | 16→256    | 50,50,50,50,50    | 38.44 / 37.58       | 0.9572 / 0.9483     | 0.0277 / 0.0297      | 74.41      | 29.68        |
> | 8→256     | 50,50,50,50,50,50 | 38.45 / 37.58       | 0.9574 / 0.9484     | 0.0277 / 0.0295      | 74.44      | 29.66        |
> | 16→256    | 16,32,31,30,29    | 33.57 / 33.19       | 0.9032 / 0.8919     | 0.0472 / 0.0487      | 68.16      | 49.21        |

---

> ### Author Response · Authors · 2025-11-21
> **Responses to reviewers: Weakness 3 and Question 2**
>
> >Subband handling is oversimplified: WRR penalizes the concatenated high-frequency bands, but no per-band analysis (LH/HL/HH individually or pairwise) is provided, despite claims about “high-frequency” components; Fig. 2a labels “high vs low”, while Haar yields LL, LH, HL, HH.
>
> >Subband granularity: WRR aggregates LH/HL/HH. What happens if only HH is penalized, or LH/HL individually, or LH+HL without HH? Any per-band sensitivity that supports the “high-frequency energy” claim beyond the aggregate.
>
> We thank the reviewer for pointing out the need for a more detailed, per-band analysis of the Haar subbands. To this end, we supplemented the ablation experiments for all subband combinations, with the results presented in Table. Several observations emerge:
>
> **No single subband dominates.**
> Applying penalties to individual subbands yields comparable reconstruction quality and robustness, indicating adversarial perturbations are not confined to a single frequency direction.
>
> **Pairwise penalties prove unstable.**
> Combining two subbands yields inconsistent results, occasionally improving robustness accuracy marginally. This suggests selective regularization of high-frequency spectral components leads to imbalanced perturbation suppression.
>
> **Simultaneously penalizing all high-frequency bands proves most stable and effective.**
> Applying WRR to the full band set (LH+HL+HH) achieves the optimal tradeoff between clean accuracy and robust accuracy across all variants. This validates our design choice: adversarial perturbations involve multi-directional components, and penalizing the concatenated high-frequency representation provides the most reliable control without overfitting to specific sub-bands.
>
> | Method        | PSNR (Clean / Adv) | SSIM (Clean / Adv) | NRMSE (Clean / Adv) | Clean Acc. | Robust Acc. |
> |---------------|---------------------|---------------------|----------------------|------------|--------------|
> | LH            | 33.29 / 32.93       | 0.8991 / 0.8877     | 0.0484 / 0.0499      | 68.21      | 47.66        |
> | HL            | 33.24 / 32.89       | 0.8984 / 0.8871     | 0.0486 / 0.0501      | 68.99      | 47.10        |
> | HH            | 33.80 / 33.37       | 0.9057 / 0.8944     | 0.0465 / 0.0481      | 68.79      | 47.85        |
> | LH + HL       | 32.65 / 33.35       | 0.8892 / 0.8777     | 0.0512 / 0.0526      | 68.51      | 47.80        |
> | LH + HH       | 32.98 / 32.74       | 0.8946 / 0.8856     | 0.0497 / 0.0508      | 68.01      | 48.41        |
> | HL + HH       | 32.94 / 32.63       | 0.8941 / 0.8827     | 0.0498 / 0.0512      | 67.79      | 48.58        |
> | LH + HH + HL  | 33.57 / 33.19       | 0.9032 / 0.8919     | 0.0472 / 0.0487      | 68.16      | 49.21        |

---

> ### Author Response · Authors · 2025-11-21
> **Responses to reviewers: Weakness 4**
>
> >Trade-off instability: In Table 2, settings that improve robust accuracy tend to hurt clean (IID) accuracy and vice-versa, and the method without a robust backbone trails strong AT in some settings; the paper does not resolve when LP+WRR is preferable.
>
> We appreciate the reviewers' insightful comments!
>
> The trade-off between clean and robust performance observed in Table 1 of our paper aligns with expectations, as this phenomenon is widely recognized in the adversarial robustness literature as an unavoidable fundamental trade-off.
>
> First, we would like to clarify that the backbones we use for different datasets follow the standard practice in the adversarial robustness literature: **WideResNet-28-10 for CIFAR-10/CIFAR-100** and **ResNet-50 for ImageNet**. The backbone description for CIFAR-10 in Table 1 was **mistakenly reported** and will be corrected in the revised version.
>
> Regarding the performance issue, our results on CIFAR-10 do not reach state-of-the-art performance primarily because WideResNet-28-10 tends to **overfit on CIFAR-10** due to its limited data size, a behavior also observed in many adversarial training (AT) methods. To address this overfitting problem, recent AT approaches typically rely on additional synthetic data to train a more robust classifier. Following this practice, we evaluated our method using the robust classifier trained with synthetic data provided by [1]. This leads to a substantial performance improvement.
>
> The results show that our approach delivers the strongest robust accuracy on CIFAR-10, which demonstrate that the effectiveness of our method and its potential for defending against adversarial attacks.
>
> [1] Jiequan Cui, Zhuotao Tian, Zhisheng Zhong, Xiaojuan Qi, Bei Yu, and Hanwang Zhang. Decoupled kullback-leibler divergence loss. Advances in Neural Information Processing Systems, 37:74461– 74486, 2024.

---

> ### Author Response · Authors · 2025-11-21
> **Responses to reviewers: Weakness 5 and Question 3, 4**
>
> >Protocol clarity: Table 1 aggregates results across datasets with different classifier backbones (ResNet-50 vs WideResNet-28-10), complicating method-to-method comparison. QTT rank settings used for headline Table-1 numbers are not explicit.
>
> >Headline settings: Which QTT rank/scale settings and WRR $\lambda$ were used for the headline Table-1 results on each dataset?
>
> >Backbone consistency: Why switch to WideResNet-28-10 on CIFAR-100 while using ResNet-50 elsewhere. Can the main table be normalized to a single backbone per dataset for fair comparison?
>
> Thank you very much for raising the question regarding the clarity of the experimental protocol!
>
> We apologize for the misunderstanding caused by my writing mistake. First, it should be clarified that the experimental results in Table 1 were obtained by strictly following the RobustBench protocol. The backbones we use for different datasets follow the standard practice in the adversarial robustness literature: **WideResNet-28-10 for CIFAR-10/CIFAR-100** and **ResNet-50 for ImageNet**. The backbone description for CIFAR-10 in Table 1 was **mistakenly reported** and will be corrected in the revised version.
>
> Regarding QTT ranks, we use a fixed rank schedule per dataset:
> CIFAR-10/100: rank = [8, 16, 10, 4], ImageNet: rank = [16, 32, 31, 30, 29].
>
> For both datasets, we set the lambda parameter to 1e-6. For Cifar10, we first upsampled images to a resolution of 64 and downsampled them to a minimum resolution of 8. For ImageNet, we upsampled images to a resolution of 256 and downsampled them to a minimum resolution of 16.

---

> > ### Comment · Reviewer_UAmz · 2025-11-26
> > **Confusion**
> >
> > > "For ImageNet, we upsampled images to a resolution of 256"
> >
> > Could you please clarify why ImageNet images need to be upsampled to 256?

---

> > > ### Author Response · Authors · 2025-11-27
> > > **Responses to reviewers: Confusion**
> > >
> > > > Could you please clarify why ImageNet images need to be upsampled to 256?
> > >
> > > We sincerely thank the reviewer for carefully reading our response and raising this question.
> > >
> > > The upsampling of ImageNet images to 256$\times$256 is required by the QTT decomposition. The QTT format builds upon mode quantization, which **decomposes the scaling dimension in powers of 2**. For example, an second-order tensor 16$\times$16 is factorized as a 8-dimensional hypercube:
> > >
> > > $(2_1\times 2_2 \times 2_3 \times 2_4) \times (2_1\times 2_2 \times 2_3 \times 2_4)$,
> > >
> > > as illustrated in [1, 2]. This quantization step is fundamental to constructing the hierarchical QTT cores.
> > >
> > > Since the native ImageNet resolution 224$\times$224 is not a power of 2, we upsample the image to 256$\times$256 to achieve a well-defined QTT decomposition and stable multi-level reconstruction.
> > >
> > > [1] Khoromskij, B. N. o(d log n)-quantics approximation of n-d tensors in high-dimensional numerical modeling. Constructive Approximation, 34:257–280, 2011.
> > >
> > > [2] Obukhov, A., Usvyatsov, M., Sakaridis, C., Schindler, K., and Van Gool, L. Tt-nf: Tensor train neural fields, 2022. https://arxiv.org/abs/2209.15529.

---

> ### Author Response · Authors · 2025-11-21
> **Responses to reviewers: Weakness 6, 7 and Question 5**
>
> >Baseline coverage and consistency: Although several AT/AP baselines are listed in Table 1, alignment with common RobustBench protocols is unclear, and baseline choices/backbones differ across datasets, reducing comparability.
>
> >Benchmark alignment: Were the AT baselines reproduced under RobustBench-style evaluation, or are they literature numbers with heterogeneous training data/regularization. If the latter, can a standardized subset be added?
>
> We apologize for the misunderstanding caused by my writing mistake. It should be clarified that the experimental results in Table 1 were obtained by strictly following the RobustBench protocol. The backbones we use for different datasets follow the standard practice in the adversarial robustness literature: **WideResNet-28-10 for CIFAR-10/CIFAR-100** and **ResNet-50 for ImageNet**.
>
> All comparison results are taken from RobustBench or from the best reported performance in the original papers. Each method is evaluated under the same experimental settings following the RobustBench protocol. Therefore, the numbers in our tables are consistent, comparable, and meaningful for reference.
>
> >Missing related work: No mention of FLC/anti-aliasing pooling literature (e.g., [1]) that is relevant to frequency-aware robustness and could inform design choices.
> References: [1] Grabinski, Julia, et al. "Frequencylowcut pooling-plug and play against catastrophic overfitting." European Conference on Computer Vision. Cham: Springer Nature Switzerland, 2022.
>
> Thank you for your suggestion. We will revise the relevant sections of our work to properly cite the FLC study.

---

> > ### Comment · Reviewer_UAmz · 2025-11-26
> >
> > Thank you for the detailed response.
> >
> > Apart from the above-mentioned question, most of my concerns have been answered.
> >
> > I have updated my rating accordingly.
> >
> > I am waiting for the response to the above question and the discussion that follows with the other reviewers before making a final recommendation.
> >
> > Best Regards

---

> > > ### Author Response · Authors · 2025-11-27
> > > **Responses to reviewers UAmz**
> > >
> > > Thank you very much for your thoughtful follow-up and for updating your rating. We sincerely appreciate the time and care you have devoted to reviewing our work and providing constructive feedback. We have responded to the new question, and we warmly welcome and look forward to any further discussion.
> > >
> > > Best regards

---

> ### Author Response · Authors · 2025-11-30
> **Responses to reviewers: The revised version has been uploaded**
>
> Dear reviewers,
>
> We sincerely thank all reviewers for their thorough reading and highly insightful comments. We have carefully considered every suggestion and have revised the manuscript accordingly. A fully updated version has now been uploaded. For ease of verification, all modifications are in blue in the revised manuscript.
>
> We very much appreciate the reviewers’ professional feedback, and we look forward to any further comments and discussion.
>
> Best regards

---

### Comment · Area_Chair_xucX · 2025-11-28

Dear Reviewers,

Thank you for your valuable time and expertise in reviewing this paper.

The authors have now submitted their rebuttal. We would appreciate it if you could review their responses and assess whether your concerns have been addressed, if you haven't done this.

Best regards,

AC

---

### Meta-Review · Area_Chair_CDt7 · 2026-01-08

**Summary:**

The paper received mixed reviews initially. The authors conducted an extensive experiments in the rebuttal, including but not limited to adding new baselines and efficiency analysis. Most of the reviewers' concerns have been resolved well. Only one of the reviewers participates in the discussion. After carefully reviewing the paper and comments, as well as the rebuttal, the AC believes there are several concerns remain unresolved, limiting the paper for acceptance.

First, the proposed ranking scheduling follows a principle while it is still heuristic. As the core contribution, heuristic approach usually demands more detailed analysis and ablations. The specific rank settings for CIFAR and ImageNet appear as "magic numbers" without a clear automatic scheme which can be easily tuned. The authors acknowledged that an adaptive scheme can be very difficult. In this case, the current ablation studies are insufficient. A denser granularity of rank variations is needed to demonstrate the selected schedule is optimal. Also we need to avoid the selected hyper-parameters from overfitting specific datasets. The generalization of the proposed scheme is not fully convincing.

Second, the proposed regularization introduced additional complexity, while the improvements seem marginal in Figure 3. The results show that $\lambda=0$  yields the best reconstruction and clean accuracy, Increasing this value will degrade the reconstruction and clean accuracy monotonically as high as 10%, in exchange for a modest gain in robust accuracy (~4%). This demonstrates that WRR might not be an effective regularizer.

Third, the state-of-the-art performance relies on the underlying robust classifier. As in Table 1, switching to a more robust classifier boost the performance. Given the complexity of the proposed pipeline, the gain from the purification module seems marginal. Also the AC verified the cited baselines and found some discrepancies between the reported numbers in Table 1, and Tables in Cui et al., and the official RobustBench leaderboard. Accuracy in baseline reporting is crucial for fair comparison. The results section needs heavy proof-reading to make it more convincing.

Therefore, the AC cannot recommend acceptance, and encourages the authors to address the above concerns via more extensive experiments and work towards an adaptive and generic framework. to improve the robustness.

**Reviewer Concerns:**

- Reviewer UAmz: most of the concerns have been well addressed and the reviewer already acknowledged that.
- Reviewer LvRB: the time measurements have been added. The defined strategy of rank selection is insufficiently addressed. The authors provided an additional ablation studies on different schedule, but not dense enough. Others are well-addressed.
- Reviewer baTF: the reviewer also mentioned the heuristic rank selection process which is not perfectly resolved. The computational cost and white-box issues have been clarified. The authors also discussed the difference between AP and AD, which is convincing.
- Reviewer QK4e: the authors clarified multiple misunderstanding of the training, and added experiments on other baselines. Efficiency issues are fully discussed. Most of the reviewer's concerns are well addressed.

**Reviewer Scores:**

Reviewer UAmz and Reviewer QK4e might upgrade the score, while Reviewer LvRB and Reviewer baTF might keep the scores unchanged.

---

### Decision · Program_Chairs · 2026-01-26

Reject